# Effects of music therapy on depression: A meta-analysis of randomized controlled trials

Qishou Tang[1], Zhaohui Huang[2], Huan Zhou[1,3], Peijie Ye [1]*

**1** Bengbu Medical University, Bengbu, Anhui, China, **2** Anhui Provincial Center for Women and Child Health, Hefei, Anhui, China, **3** National Drug Clinical Trial Institution, The First Affiliated Hospital of Bengbu Medical University, Bengbu, Anhui, China

\* 459695783@qq.com

## Abstract

### Background

We aimed to determine and compare the effects of music therapy and music medicine on depression, and explore the potential factors associated with the effect.

### Methods

PubMed (MEDLINE), Ovid-Embase, the Cochrane Central Register of Controlled Trials, EMBASE, Web of Science, and Clinical Evidence were searched to identify studies evaluating the effectiveness of music-based intervention on depression from inception to May 2020. Standardized mean differences (SMDs) were estimated with random-effect model and fixed-effect model.

### Results

A total of 55 RCTs were included in our meta-analysis. Music therapy exhibited a significant reduction in depressive symptom (SMD = −0.66; 95% CI = -0.86 to -0.46; $P<0.001$) compared with the control group; while, music medicine exhibited a stronger effect in reducing depressive symptom (SMD = −1.33; 95% CI = -1.96 to -0.70; $P<0.001$). Among the specific music therapy methods, recreative music therapy (SMD = -1.41; 95% CI = -2.63 to -0.20; $P<0.001$), guided imagery and music (SMD = -1.08; 95% CI = -1.72 to -0.43; $P<0.001$), music-assisted relaxation (SMD = -0.81; 95% CI = -1.24 to -0.38; $P<0.001$), music and imagery (SMD = -0.38; 95% CI = -0.81 to 0.06; $P = 0.312$), improvisational music therapy (SMD = -0.27; 95% CI = -0.49 to -0.05; $P = 0.001$), music and discuss (SMD = -0.26; 95% CI = -1.12 to 0.60; $P = 0.225$) exhibited a different effect respectively. Music therapy and music medicine both exhibited a stronger effects of short and medium length compared with long intervention periods.

### Conclusions

A different effect of music therapy and music medicine on depression was observed in our present meta-analysis, and the effect might be affected by the therapy process.

**Data Availability Statement:** All relevant data are within the manuscript and its Supporting Information files.

**Funding:** The Key Project of University Humanities and Social Science Research in Anhui Province (SK2017A0191) was granted by Education

Department of Anhui Province; the Research Project of Anhui Province Social Science Innovation Development (2018XF155) was granted by Anhui Provincial Federation of Social Sciences; the Ministry of Education Humanities and Social Sciences Research Youth fund Project (17YJC840033) was granted by Ministry of Education of the People's Republic of China. These funders had a role in study design, text editing, interpretation of results, decision to publish and preparation of the manuscript.

**Competing interests:** The authors have declared that no competing interests exist.

## Introduction

Depression was reported to be a common mental disorders and affected more than 300 million people worldwide, and long-lasting depression with moderate or severe intensity may result in serious health problems [1]. Depression has become the leading causes of disability worldwide according to the recent World Health Organization (WHO) report. Even worse, depression was closely associated with suicide and became the second leading cause of death, and nearly 800 000 die of depression every year worldwide [1, 2]. Although it is known that treatments for depression, more than 3/4 of people in low and middle-income income countries receive no treatment due to a lack of medical resources and the social stigma of mental disorders [3]. Considering the continuously increased disease burden of depression, a convenient effective therapeutic measures was needed at community level.

Music-based interventions is an important nonpharmacological intervention used in the treatment of psychiatric and behavioral disorders, and the obvious curative effect on depression has been observed. Prior meta-analyses have reported an obvious effect of music therapy on improving depression [4, 5]. Today, it is widely accepted that the music-based interventions are divided into two major categories, namely music therapy and music medicine. According to the American Music Therapy Association (AMTA), "music therapy is the clinical and evidence-based use of music interventions to accomplish individualized goals within a therapeutic relationship by a credentialed professional who has completed an approved music therapy program" [6]. Therefore, music therapy is an established health profession in which music is used within a therapeutic relationship to address physical, emotional, cognitive, and social needs of individuals, and includes the triad of music, clients and qualified music therapists. While, music medicine is defined as mainly listening to prerecorded music provided by medical personnel or rarely listening to live music. In other words, music medicine aims to use music like medicines. It is often managed by a medical professional other than a music therapist, and it doesn't need a therapeutic relationship with the patients. Therefore, the essential difference between music therapy and music medicine is about whether a therapeutic relationship is developed between a trained music therapist and the client [7–9]. In the context of the clear distinction between these two major categories, it is clear that to evaluate the effects of music therapy and other music-based intervention studies on depression can be misleading. While, the distinction was not always clear in most of prior papers, and no meta-analysis comparing the effects of music therapy and music medicine was conducted. Just a few studies made a comparison of music-based interventions on psychological outcomes between music therapy and music medicine. We aimed to (1) compare the effect between music therapy and music medicine on depression; (2) compare the effect between different specific methods used in music therapy; (3) compare the effect of music-based interventions on depression among different population [7, 8].

## Materials and methods

### Search strategy and selection criteria

PubMed (MEDLINE), Ovid-Embase, the Cochrane Central Register of Controlled Trials, EMBASE, Web of Science, and Clinical Evidence were searched to identify studies assessing the effectiveness of music therapy on depression from inception to May 2020. The combination of "depress*" and "music*" was used to search potential papers from these databases. Besides searching for electronic databases, we also searched potential papers from the reference lists of included papers, relevant reviews, and previous meta-analyses. The criteria for selecting the papers were as follows:(1) randomised or quasi-randomised controlled trials; (2)

music therapy at a hospital or community, whereas the control group not receiving any type of music therapy; (3) depression rating scale was used. The exclusive criteria were as follows: (1) non-human studies; (2) studies with a very small sample size (n<20); (3) studies not providing usable data (including sample size, mean, standard deviation, etc.); (4) reviews, letters, protocols, etc. Two authors independently (YPJ, HZH) searched and screened the relevant papers. EndNote X7 software was utilized to delete the duplicates. The titles and abstracts of all searched papers were checked for eligibility. The relevant papers were selected, and then the full-text papers were subsequently assessed by the same two authors. In the last, a panel meeting was convened for resolving the disagreements about the inclusion of the papers.

## Data extraction

We developed a data abstraction form to extract the useful data: (1) the characteristics of papers (authors, publish year, country); (2) the characteristics of participators (sample size, mean age, sex ratio, pre-treatment diagnosis, study period); (3) study design (random allocation, allocation concealment, masking, selection process of participators, loss to follow-up); (4) music therapy process (music therapy method, music therapy period, music therapy frequency, minutes per session, and the treatment measures in the control group); (5) outcome measures (depression score). Two authors independently (TQS, ZH) abstracted the data, and disagreements were resolved by discussing with the third author (YPJ).

## Assessment of risk of bias in included studies

Two authors independently (TQS, ZH) assessed the risk of bias of included studies using Cochrane Collaboration's risk of bias assessment tool, and disagreements were resolved by discussing with the third author (YPJ) [10].

## Music therapy and music medicine

Music Therapy is defined as the clinical and evidence-based use of music interventions to accomplish individualed goals within a therapeutic relationship by a credentialed professional who has completed an approved music therapy program. Music medicine is defined as mainly listening to prerecorded music provided by medical personnel or rarely listening to live music. In other words, music medicine aims to use music like medicines.

Music therapy mainly divided into active music therapy and receptive music therapy. Active music therapy, including improvisational, re-creative, and compositional, is defined as playing musical instruments, singing, improvisation, and lyrics of adaptation. Receptive music therapy, including music-assisted relaxation, music and imagery, guided imagery and music, lyrics analysis, and so on, is defined as music listening, lyrics analysis, and drawing with musing. In other words, in active methods participants are making music, and in receptive music therapy participants are receiving music [6, 7, 9, 11–13].

## Evaluation of depression

Depression was evaluated by the common psychological scales, including Beck Depression Inventory (BDI), Children's Depression Inventory (CDI), Center for Epidemiologic Studies Depression (CES-D), Cornell Scale (CS), Depression Mood Self-Report Inventory for Adolescence (DMSRIA), Geriatric Depression Scale-15 (GDS-15); Geriatric Depression Scale-30 (GDS-30), Hospital Anxiety and Depression Scale (HADS), Hamilton Rating Scale for Depression (HRSD/HAMD), Montgomery-sberg Depression Rating Scale (MADRS), Patient

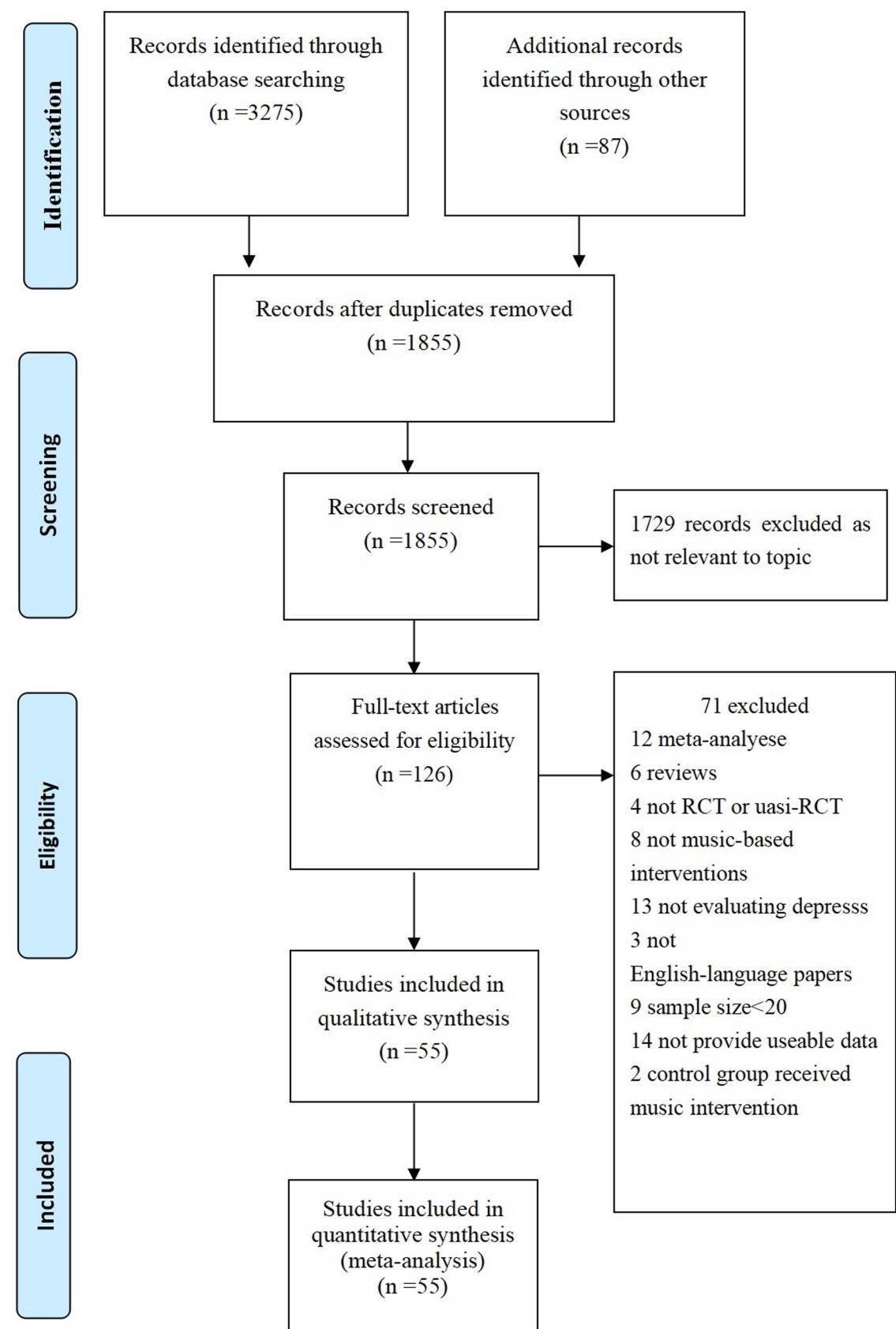

**Fig 1. Prisma 2009 flow diagram literature search and study selection.** PRISMA diagram showing the different steps of systematic review, starting from literature search to study selection and exclusion. At each step, the reasons for exclusion are indicated. Doi: 10.1371/journal.pone.0052562.g001.

Reported Outcomes Measurement Information System (PROMIS), Self-Rating Depression Scale (SDS), Short Version of Profile of Mood States (SV-POMS).

## Statistical analysis

The pooled effect were estimated by using the standardized mean differences (SMDs) and its 95% confidence interval (95% CI) due to the different depression rate scales were used in the included papers. Heterogeneity between studies was assessed by I-square ($I^2$) and Q-statistic (P<0.10), and a high $I^2$ (>50%) was recognized as heterogeneity and a random-effect model was used [14–16]. We performed subgroup analyses and meta-regression analyses to study the potential heterogeneity between studies. The subgroup variables included music intervention categories (music therapy and music medicine), music therapy methods (active music therapy, receptive music therapy), specific receptive music therapy methods (music-assisted relaxation, music and imagery, and guided imagery and music (Bonny Method), specific active music therapy methods (recreative music therapy and improvisational music therapy), music therapy mode (group therapy, individual therapy), music therapy period (weeks) (2–4, 5–12, ≥13), music therapy frequency (once weekly, twice weekly, ≥3 times weekly), total music therapy sessions (1–4, 5–8, 9–12, 13–16, >16), time per session (minutes) (15–40, 41–60, >60), inpatient settings (secure [locked] unit at a mental health facility versus outpatient settings), sample size (20–50, ≥50 and <100, ≥100), female predominance(>80%) (no, yes), mean age (years) (<50, 50–65, >65), country having music therapy profession (no, yes), pre-treatment diagnosis (mental health, depression, severe mental disease/psychiatric disorder). We also performed sensitivity analyses to test the robustness of the results by re-estimating the pooled effects using fixed effect model, using trim and fill analysis, excluding the paper without information on music therapy, excluding the papers with more high biases, excluding the papers with small sample size (20< n<30), excluding the papers using an infrequently used scale, excluding the studies focused on the people with a severe mental disease. We investigated the publication biases by a funnel plot as well as Egger's linear regression test [17]. The analyses were performed using Stata, version 11.0. All P-values were two-sided. A P-value of less than 0.05 was considered to be statistically significant.

## Results

### Characteristics of the eligible studies

Fig 1 depicts the study profile, and a total of 55 RCTs were included in our meta-analysis [18–72]. Of the 55 studies, 10 studies from America, 22 studies from Europe, 22 studies from Asia, and 1 study from Australia. The mean age of the participators ranged from 12 to 86; the sample size ranged from 20 to 242. A total of 16 different scales were used to evaluate the depression level of the participators. A total of 25 studies were conducted in impatient setting and 28 studies were in outpatients setting; 32 used a certified music therapist, 15 not used a certified music therapist (for example researcher, nurse), and 10 not reported relevent information. A total of 16 different depression rating scales were used in the included studies, and HADS, GDS, and BDI were the most frequently used scales (Table 1).

Of the 55 studies, only 2 studies had high risks of selection bias, and almost all of the included studies had high risks of performance bias (Fig 2).

**Table 1. Characteristics of clinical trials included in this meta-analysis.**

| Studies | Country | Ample size | Mean age (SD) | Pre-intervention diagnosis | Music intervention method (total) | Intervenor or therapist | Intervention description | Control group | Outcome Measures |
|---|---|---|---|---|---|---|---|---|---|
| Biasutti et al., 2019 | Italy | N = 45, Female = 29 | 84.6 (7.17) | Healthy or with cognitive impairment | Active music therapy (improvisational music therapy) | Certified music therapist | Twice weekly (70 min/ session) for 6 weeks | 45-minute gymnastic activities | GDS-15 |
| Burrai et al., [48] | Italy | N = 159, Female = 124 | 73.05 (11.5) | Heart failure | Music medicine | Researchers | Once daily (30 min/session) for 36 weeks | Standard HF treatment | HADS |
| Burrai et al., [49] | Italy | N = 24, Female = 9 | 62.3 (2.8) | End-stage kidney disease | Music medicine | Nurse | Once daily (15 min/session) for 2 weeks | Standard hemodialysis | HADS |
| Chan et al., 2009 | Hong Kong China | N = 47, Female = 26 | >60 | No mental illness | Music medicine | Researchers | Once weekly (30 min/ session) for 4 weeks | Without any intervention | GDS-30 |
| Chan et al., 2010 | Hong Kong China | N = 42, Female = 23 | >60 | No mental illness | Music medicine | Researchers | Once weekly (45 min/ session) for 4 weeks | Without any intervention | GDS-15 |
| Chan et al., 2012 | Singapore | N = 50, Female = 32 | >55 | No mental illness | Music medicine | Researchers | Once weekly (30 min/ session) for 8 weeks | Without any intervention | GDS-15 |
| Chen et al., 2015 | Taiwan China | N = 71, Female = 69 | 18.5 | Depressive disorder | Music medicine | Researchers | Twice weekly (40 min/ session) for 10 weeks | Without any intervention | DMSRIA |
| Chen et al., 2018 | China | N = 52, Female = 52 | - | Breast cancer | Receptive music therapy | Certified music therapist | Once weekly (60 min/ session) for 8 weeks | Standard care | HADS |
| Chen et al., 2019 | Taiwan China | N = 65, Female = 56 | 72.7 (5.97) | No mental illness | Active music therapy (improvisational music therapy) | Not reported | Twice weekly (40 min/ session) for 10 weeks | No music therapy | BDI |
| Cheung et al., 2019 | Hong Kong, China | N = 60, Female = 25 | 13.2 (3.27) | Pediatric brain tumor with a significant level of depression | Active music therapy (recreative music therapy) | Certified music therapist | Once weekly (45 min/ session) for 52 weeks | No music therapy | CES-D |
| Chirico et al., 2020 | Italy | N = 64, Female = 64 | 55.95 (5.92) | Breast cancer | Receptive music therapy | Certified music therapist | 20 min/session | Standard care | SV-POMS |
| Choi et al., 2008 | Korea | N = 26, Female = 14 | 36.15 (10.2) | Psychiatric disorder | Active music therapy (recreative music therapy) | Certified music therapist | Once-two weekly (60 min/session) for 12 weeks | Routine care | BDI |
| Chu et al., 2014 | Taiwan, China | N = 100, Female = 53 | 82(6.8) | Dementia | Active music therapy (improvisational music therapy) | Certified music therapist | Twice weekly (30 min/ session) for 6 weeks | Standard care | CS |
| Cooke et al., 2010 | Australia | N = 47, Female = 33 | >65 | Dementia | Active music therapy (improvisational music therapy) | Musicians | Thrice weekly (40 min/ session) for 8 weeks | Educational/ entertainment activities | GDS |
| Erkkilä et al., 2011 | Finland | N = 79, Female = 62 | 35.6 (9.75) | Depression disorder | Active music therapy (improvisational music therapy) | Certified music therapist | Twice weekly (60 min/ session) for 12 weeks | Standard treatment | MADRS |

(*Continued*)

**Table 1.** (Continued)

| Studies | Country | Ample size | Mean age (SD) | Pre-intervention diagnosis | Music intervention method (total) | Intervenor or therapist | Intervention description | Control group | Outcome Measures |
|---|---|---|---|---|---|---|---|---|---|
| Fancourt et al., 2019 | UK | N = 62, Female = 48 | 54.5 (14.5) | Cancer carers | Active music therapy (improvisational music therapy) | Certified music therapist | Once weekly (90 min/session) for 12 weeks | No music therapy | HADS |
| Gok Ugur et al., 2017 | Turkey | N = 64, Female = 22 | 76.35 (7.88) | No mental illness | Receptive music therapy (music and imagery) | Certified music therapist | Three days in a week for 8 weeks | No music therapy | GDS-15 |
| Guétin et al., 2009 | France | N = 30, Female = 22 | 86(5.6) | Moderate stages of Alzheimer's disease | Receptive music therapy (music-assisted relaxation) | Certified music therapist | Once weekly (20 min/session) for 16 weeks | Educational/entertainment activities | GDS-30 |
| Hanser et al., 1994 | USA | N = 30, Female = 23 | 67.9 | Depressive disorder | Receptive music therapy (guided imagery and music) | Certified music therapist | Once weekly (1 h/session; 20 min/session) for 8 weeks | No music therapy | GDS |
| Hars et al., 2014 | Switzerland | N = 134, Female = 129 | 75(7) | No mental illness | Music medicine | Not reported | Once weekly (1 h/session) for 26 weeks | No music therapy | HADS |
| Liao et al., 2018 | China | N = 107, Female = 66 | 71.79 (7.71) | Mild to moderate depressive symptoms | Music medicine | Not reported | Once weekly (50 min/session) for 12 weeks | Routine health education | GDS-30 |
| Low et al., 2020 | USA | N = 43, Female = 33 | 50.07 (5.48) | Chronic pain | Active+receptive music therapy | Certified music therapist | Once weekly (90 min/session) for 12 weeks | Standard care | PROMIS |
| Mahendran et al., 2018 | Singapore | N = 68, Female = 56 | 71.1 (5.3) | Mild cognitive impairment | Receptive music therapy (guided imagery and music) | Certified music therapist | Once weekly for 3 months, then fortnightly for 36 weeks. | No music therapy | GDS-15 |
| Park et al., 2015 | South Korea | N = 29, Female = 16 | 8.17 (1.47) | No mental illness | Active music therapy (improvisational music therapy) | Music therapist | Once weekly (120 min/session) for 15 weeks | Educational creative movement program | CDI |
| Pérez-Ros et al., 2019 | Spain | N = 119, Female = 61 | 80.52 (7.44) | No mental illness | Active music therapy (improvisational music therapy) | Physiotherapists | 5 times weekly (60 min/session) for 8 weeks | No music therapy | CS |
| Ploukou et al., 2018 | Greece | N = 48, Female = 46 | - | Oncology nurses without diseases | Music medicine | Not reported | Once weekly (60 min/session) for 4 weeks | No music therapy | HADS |
| Ribeiro et al., 2018 | Brazil | N = 21, Female = 21 | 22.5 (6.5) | Mothers of preterm | Receptive music therapy (music and discuss) | Certified music therapist | Once weekly (30–40 min/session) for 7–9 weeks | No music therapy | BDI |
| Sigurdardóttir et al., 2019 | Denmark | N = 38, Female = 25 | 25.4 | Mild and moderate depression | Music medicine | Not reported | Twice weekly (20 min/session) for 4 weeks | No music therapy | HRSD-6, HRSD-17 |
| Toccafondi et al., 2018 | Italy | N = 242, Female = 147 | >18 | Cancer | Receptive music therapy | Certified music therapist | Once weekly | Standard care | HADS |

(*Continued*)

**Table 1.** (Continued)

| Studies | Country | Ample size | Mean age (SD) | Pre-intervention diagnosis | Music intervention method (total) | Intervenor or therapist | Intervention description | Control group | Outcome Measures |
|---|---|---|---|---|---|---|---|---|---|
| Trimmer et al., 2018 | Canada | N = 28, Female = 15 | 43(13.8) | Depression and anxiety | Active music therapy (recreative music therapy) | Not reported | Once weekly (90 min/session) for 9 weeks | Treatment as usual | HADS |
| Volpe et al., 2018 | Italy | N = 106, Female = 106 | 43.83 (12.7) | Psychosis | Active music therapy (improvisational music therapy) | Certified music therapist | Twice daily (60 min/session) for 6 weeks | Standard drug treatment | HADS |
| Wu et al., 2019 | China | N = 60, Female = 60 | 36.2 (9.47) | Methamphetamine use disorder | Active+receptive music therapy | Certified music therapist | Once weekly (90 min/session) for 13 weeks | Standard treatment | SDS |
| Albornoz et al., 2011 | Venezuela | N = 24, Female = 0 | 16–60 | Depressed adults with substance abuse | Active music therapy (improvisational music therapy) | Therapist | Once weekly (120 min/session) for 12 weeks | Standard treatment | BDI, HRSD |
| Hendricks et al., 1999 | USA | N = 20 | 14–15 | Depression | Active+receptive music therapy | Therapist | Once weekly for 8 weeks | Individual psychotherapy | BDI |
| Hendricks et al., 2001 | USA | N = 63 | 12–18 | Depression | Music medicine | counsellor-researcher | Once weekly (60 min/session) for 12 weeks | Cognitive-based psychotherapy | BDI |
| Radulovic et al., 1996 | Serbia | N = 60 | 21–62 (40) | Depression | Receptive music therapy | Therapist | Twice weekly (20 min/session) for 6 weeks | Treatment as usual | BDI |
| Zerhusen et al., 1995 | USA | N = 60 | 70–82 (77) | Moderate to severe depression | Music medicine | Not reported | Twice weekly (30 min/session) for 10 weeks | psychological therapy or treatment as usual | BDI |
| Chang et al., 2008 | Taiwan China | N = 236, Female = 236 | 22-41 (30.03) | Pregnant women | Music medicine | Music faculty members | Once a day (30 min/session) for 2 weeks | General prenatal care | EPDS |
| Chen et al., 2020 | Taiwan China | N = 100 Female = 100 | 30.19 (9.50) | Beast cancer undergoing chemotherapy. | Receptive music therapy | Trained music therapist | Once weekly (45 min/session) for 3 weeks | Routine nursing care | HADS |
| Chen et al., 2016 | China | N = 200, Female = 0 | 35.5 (9.75) | Prisoners with mild depression; | Active+receptive music therapy, including music and imagery, improvisation, and song writing | Music therapist | Twice weekly (90 min/session) for 3 weeks | Standard care | BDI |
| Esfandiari et al., 2014 | Iran | N = 30, Female = 30 | Not reported | Severe depressive disorder | Music medicine | not reported | 90 min/session | Standard care | BDI |
| Fancourt et al., 2016 | UK | N = 45, Female = 37 | 53.54 (13.85) | Mental health service users | Music medicine | Professional drummer | Once weekly (90 min/session) for 10 weeks | Without any intervention | HADS |
| Giovagnoli et al., 2017 | Italy | N = 39, Female = 24 | 73.64 (7.11) | Mild to moderate Alzheimer's disease | Active music therapy (Improvisational music therapy) | Music therapist | Twice weekly (45 min/session) for 12 weeks | Cognitive training or neuroeducation | BDI |
| Harmat et al., 2008 | Hungary | N = 94, Female = 73 | 22.6 (2.83) | Seep complaints | Music medicine | Investigators | Once a day (45 min/session) for 3 weeks | listening to an audiobook or no intervention | BDI |

(Continued)

**Table 1.** (Continued)

| Studies | Country | Ample size | Mean age (SD) | Pre-intervention diagnosis | Music intervention method (total) | Intervenor or therapist | Intervention description | Control group | Outcome Measures |
|---|---|---|---|---|---|---|---|---|---|
| Koelsch et al., 2010 | Germany | N = 154, Female = 78 | 24.6 | No disease | Active music therapy | Music therapist | Not reported | Individual psychotherapy | POMS |
| Liao et al., 2018 | China | N = 60, Female = 30 | 61.82 (13.20) | Cancer | Receptive music therapy+muscle relaxation training | not reported | Once a day (40 min/session) for 8 weeks | Muscle relaxation training | HADS |
| Lu et al., 2013 | Taiwan China | N = 80, Female = 21 | 52.02 (7.64) | Schizophrenia | Active music therapy+receptive music therapy | Trained research assistant | Twice weekly (60 min/session) for 5 weeks | Usual care | CDSS |
| Mahendran et al., 2018 | Singapore | N = 68, Female = 56 | 71.1 (5.05) | Mild cognitive impairment | Receptive music therapy | Music therapist | Weekly in the first 3 months, then fortnightly for 6 months. | Standard care without any intervention | GDS-15 |
| Mondanaro et al., 2017 | Italy | N = 60, Female = 35 | 48.20 (4.49) | Patients after spine surgery | Active music therapy (improvisational music therapy) | Music therapist | 30-minute music therapy session during an 8-hour period within 72 hours after surgery | Standard care without any intervention | HADS |
| Nwebube et al., 2017 | UK | N = 36, Female = 36 | Not reported | Pregnant women | Music medicine | Investigators | Once a day (20 min/session) for 12 weeks | Standard care without any intervention | EPDS |
| Porter et al., 2017 | Northern Ireland | N = 184, Female = 73 | 12.7 (2.5) | Adolescents with behavioural and emotional problems | Active music therapy (improvisational music therapy) | Music therapist | Once weekly (30 min/session) for 13 weeks | Usual care | CES-D |
| Raglio et al., 2016 | Italy | N = 30, Female = 17 | 64 (10.97) | Amyotrophic lateral sclerosis | Active music therapy | Music therapist | Three times weekly (30 min/session) for 4 weeks | Standard care | HADS |
| Torres, et al., 2018 | Spanish | N = 70, Female = 70 | 35-65 (51.3) | Fibromyalgia | Receptive music therapy | Music therapist | Once weekly (120 min/session) for 12 weeks | Without any additional service | ST/DEP |
| Wang et al., 2011 | China | N = 80, Female = 21 | 19.35 (1.68) | Student | Receptive music therapy | Not reported | Not reported | Without any additional service | SDS |
| Yap et al., 2017 | Singapore | N = 31, Female = 29 | 74.65 (6.4) | Elderly people | Active music therapy (improvisational music therapy) | Experienced instructors | Once weekly (60 min/session) for 11 weeks | Without any intervention | GDS |

Note: BDI = Beck Depression Inventory; CDI = Children's Depression Inventory; CDSS = depression scale for schizophrenia; CES-D = Center for Epidemiologic Studies Depression; CS = Cornell Scale; DMSRIA = Depression Mood Self-Report Inventory for Adolescence; EPDS = Edinburgh Postnatal Depression Scale; GDS-15 = Geriatric Depression Scale-15; GDS-30 = Geriatric Depression Scale-30; HADS = Hospital Anxiety and Depression Scale; HRSD (HAMD) = Hamilton Rating Scale for Depression; MADRS = Montgomery-sberg Depression Rating Scale; PROMIS = Patient Reported Outcomes Measurement Information System; SDS = Self-Rating Depression Scale; State-Trait Depression Questionnaire = ST/DEP; SV-POMS = short version of Profile of Mood States; NA = not available.

## The overall effects of music therapy

Of the included 55 studies, 39 studies evaluated the music therapy, 17 evaluated the music medicine. Using a random-effects model, music therapy was associated with a significant

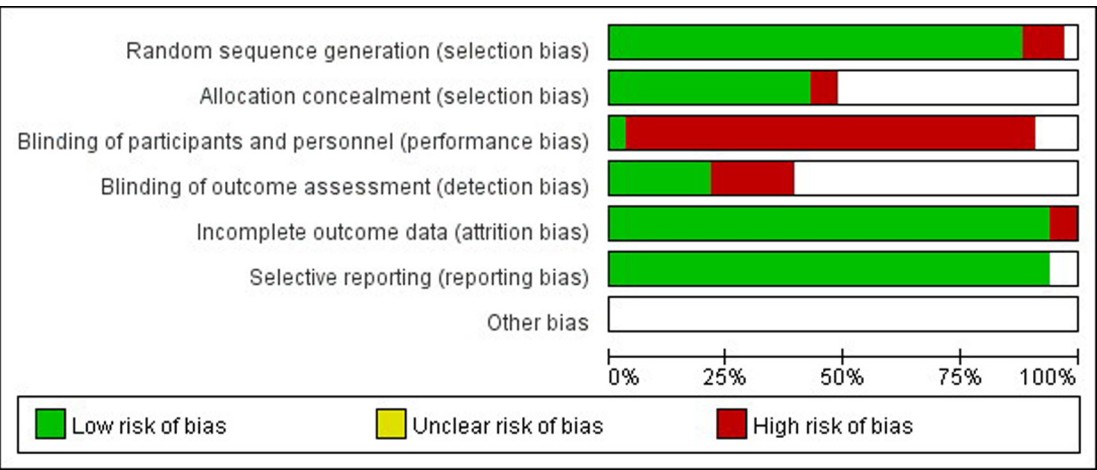

**Fig 2. Risk-of-bias graph and risk.**

reduction in depressive symptoms with a moderate-sized mean effect (SMD = −0.66; 95% CI = -0.86 to -0.46; $P<0.001$), with a high heterogeneity across studies ($I^2$ = 83%, $P<0.001$); while, music medicine exhibited a stronger effect in reducing depressive symptom (SMD = −1.33; 95% CI = -1.96 to -0.70; $P<0.001$) (Fig 3).

Twenty studies evaluated the active music therapy using a random-effects model, and a moderate-sized mean effect (SMD = −0.57; 95% CI = -0.90 to -0.25; $P<0.001$) was observed with a high heterogeneity across studies ($I^2$ = 86.3%, $P<0.001$). Fourteen studies evaluated the receptive music therapy using a random-effects model, and a moderate-sized mean effect (SMD = −0.73; 95% CI = -1.01 to -0.44; $P<0.001$) was observed with a high heterogeneity across studies ($I^2$ = 76.3%, $P<0.001$). Five studies evaluated the combined effect of active and receptive music therapy using a random-effects model, and a moderate-sized mean effect (SMD = −0.88; 95% CI = -1.32 to -0.44; $P<0.001$) was observed with a high heterogeneity across studies ($I^2$ = 70.5%, $P<0.001$) (Fig 4).

Among specific music therapy methods, recreative music therapy (SMD = -1.41; 95% CI = -2.63 to -0.20; $P<0.001$), guided imagery and music (SMD = -1.08; 95% CI = -1.72 to -0.43; $P<0.001$), music-assisted relaxation (SMD = -0.81; 95% CI = -1.24 to -0.38; $P<0.001$), music and imagery (SMD = -0.38; 95% CI = -0.81 to 0.06; $P$ = 0.312), improvisational music therapy (SMD = -0.27; 95% CI = -0.49 to -0.05; $P$ = 0.001), and music and discuss (SMD = -0.26; 95% CI = -1.12 to 0.60; $P$ = 0.225) exhibited a different effect respectively (Fig 5).

## Sub-group analyses and meta-regression analyses

We performed sub-group analyses and meta-regression analyses to study the homogeneity. We found that music therapy yielded a superior effect on reducing depression in the studies with a small sample size (20–50), with a mean age of 50–65 years old, with medium intervention frequency (<3 times weekly), with more minutes per session (>60 minutes). We also found that music therapy exhibited a superior effect on reducing depression among people with severe mental disease /psychiatric disorder and depression compared with mental health people. While, whether the country have the music therapy profession, whether the study used group therapy or individual therapy, whether the study was in the outpatients setting or the inpatient setting, and whether the study used a certified music therapist all did not exhibit a

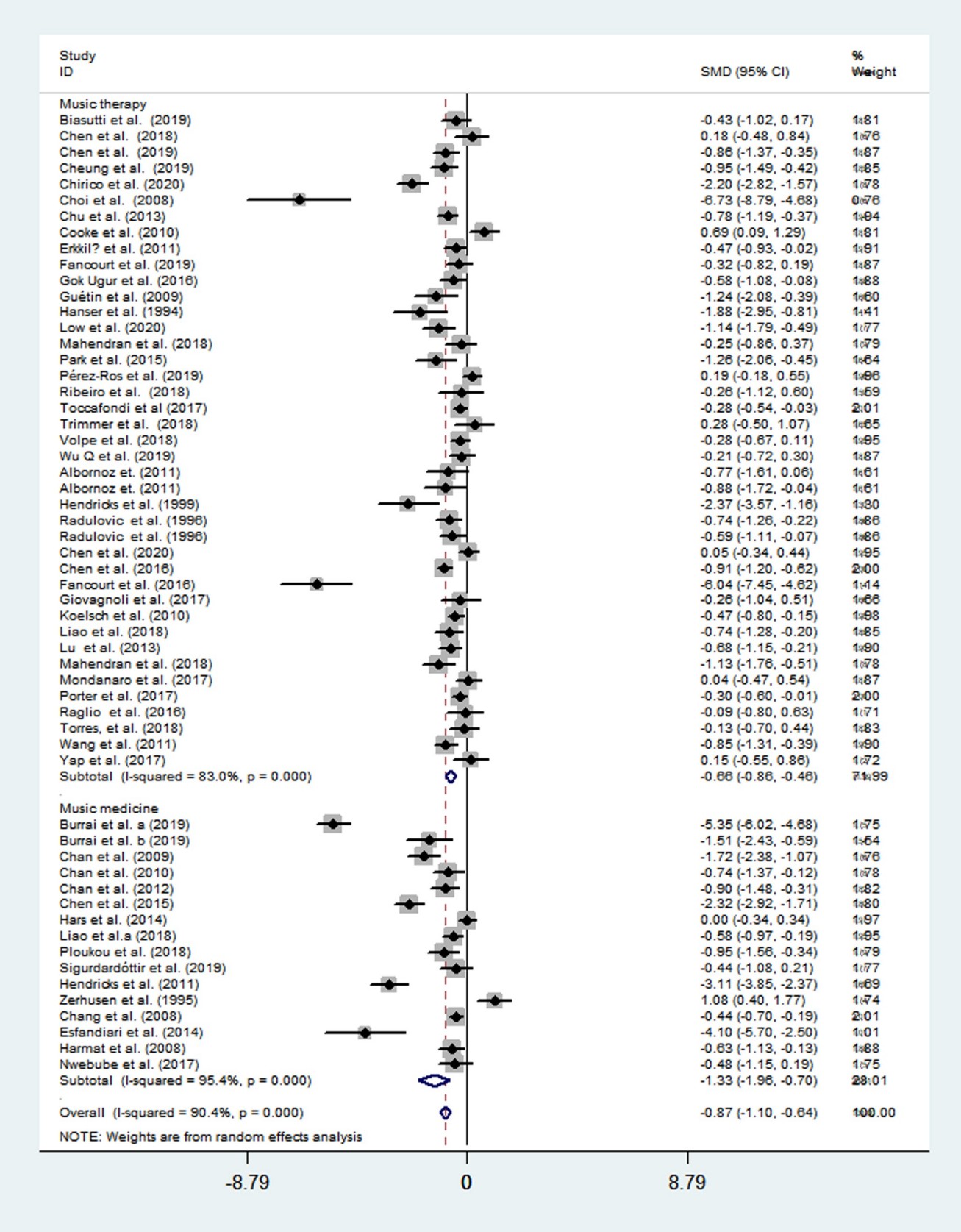

**Fig 3. Effects of music therapy and music medicine to reduce depression.**

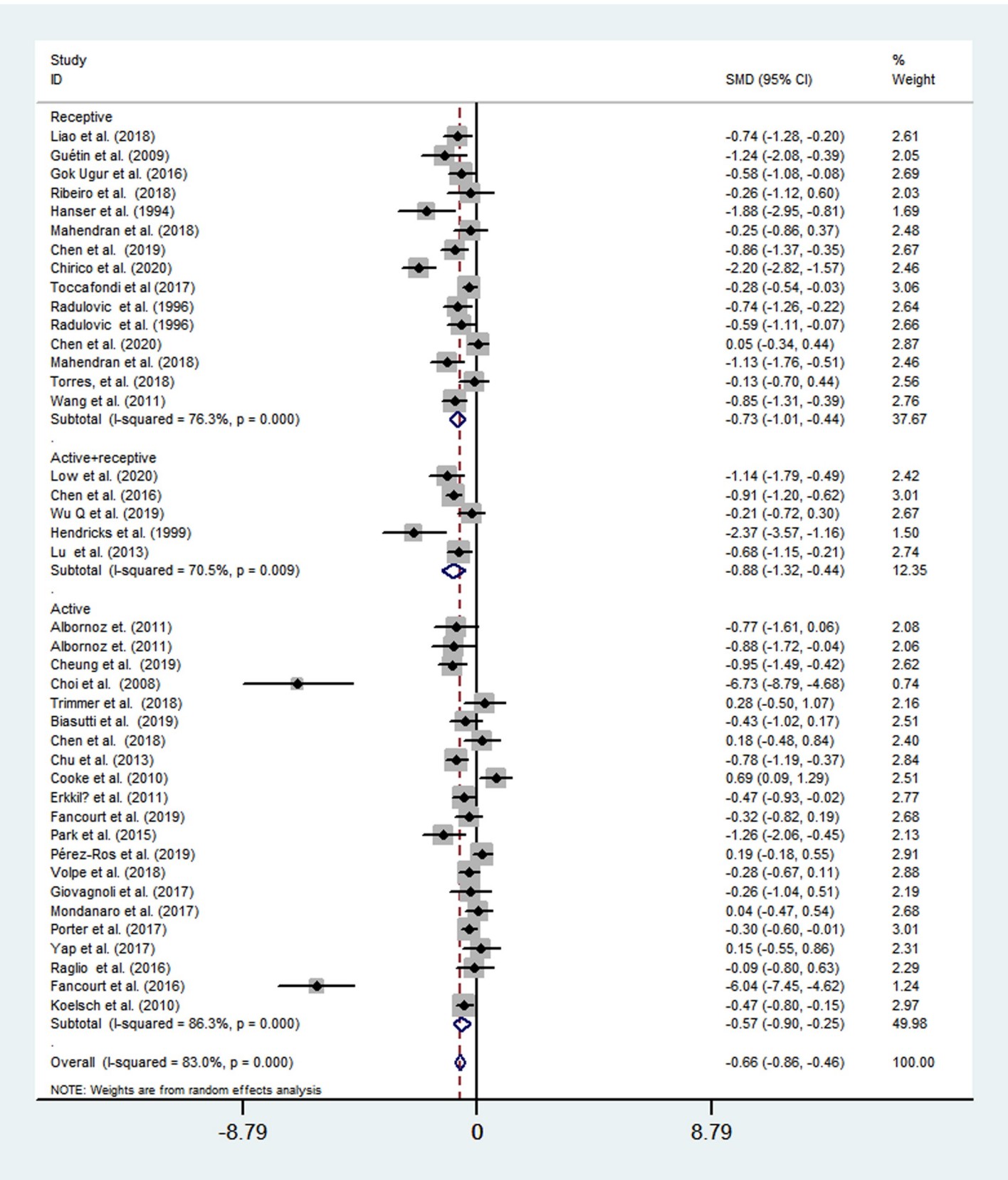

**Fig 4. Effects of active music therapy, receptive therapy, and music therapy+receptive therapy to reduce depression.**

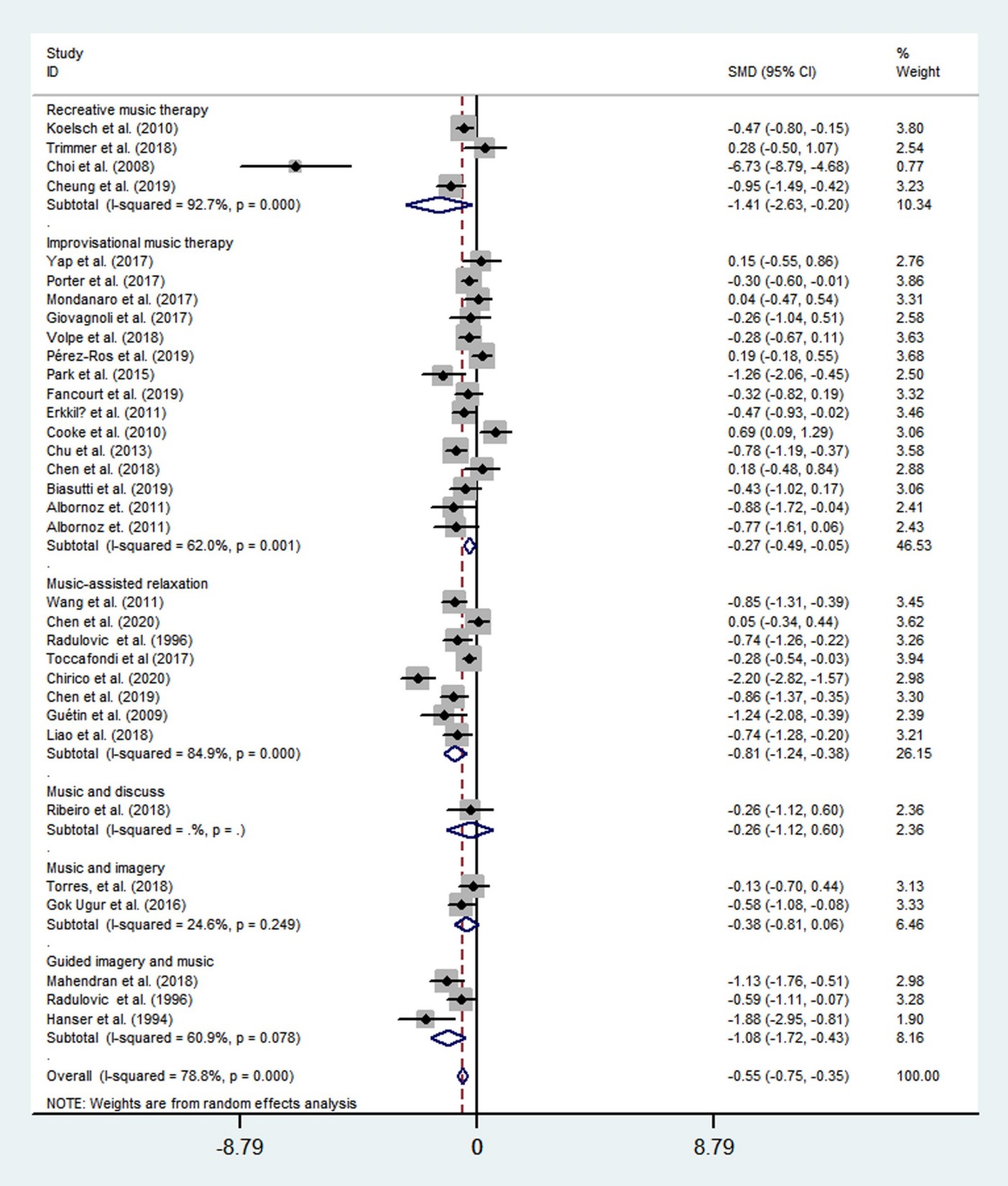

**Fig 5. Effects of specific music therapy method to reduce depression.**

**Table 2. Subgroup analyses of music-based intervention to reduce depression.**

| Subgroups | Music therapy | | | | | Music medicine | | | | |
|---|---|---|---|---|---|---|---|---|---|---|
| | Trials number | Effects | | Heterogeneity | | Trials number | Effects | | Heterogeneity | |
| | | SMD (95%CI) | *P* | *I²(%)* | *P* | | SMD (95%CI) | *P* | *I²(%)* | *P* |
| Sample size | | | | | | | | | | |
| 20–50 | 16 | -1.24(-2.08, -0.39) | <0.001 | 143.19 | <0.001 | 7 | -1.21(-1.79, -0.62) | <0.001 | 26.30 | <0.001 |
| ≥50, <100 | 17 | -0.62(-0.84, -0.38) | <0.001 | 51.58 | <0.001 | 5 | -1.17(-2.45, 0.11) | 0.073 | 86.86 | <0.001 |
| ≥100 | 8 | -0.36(-0.60, -0.11) | 0.005 | 31.33 | <0.001 | 4 | -1.56(-3.10, -0.02) | 0.047 | 206.10 | <0.001 |
| Female predominance (>80%) | | | | | | | | | | |
| Yes | 13 | -0.73(-1.23, -0.22) | 0.005 | 112.85 | <0.001 | 8 | -1.71(-2.76, -0.65) | 0.001 | 247.54 | <0.001 |
| No | 24 | -0.58(-0.81, -036) | <0.001 | 109.59 | <0.001 | 6 | -0.93(-1.32, -0.54) | <0.001 | 12.51 | 0.028 |
| Mean age (years) | | | | | | | | | | |
| <50 | 20 | -0.6(-0.85, -0.35) | <0.001 | 84.50 | <0.001 | 5 | -1.36(-2.30, -0.41) | 0.005 | 69.99 | <0.001 |
| 50–65 | 7 | -1.43(-2.28, -0.58) | 0.001 | 78.58 | <0.001 | 2 | -1.10(-1.66, -0.53) | <0.001 | 1.22 | <0.001 |
| >65 | 12 | -0.48(-0.84, -0.13) | 0.008 | 48.47 | <0.001 | 6 | -1.21(-2.66, 0.24) | 0.102 | 237.19 | <0.001 |
| Pre-treatment diagnosis | | | | | | | | | | |
| Mental health | 23 | -0.58(-0.85, -0.32) | <0.001 | 141.40 | <0.001 | 10 | -1.26(-2.04, -0.47) | 0.002 | 218.03 | <0.001 |
| Depression | 9 | -0.79(-1.13, -0.46) | <0.001 | 20.83 | <0.001 | 6 | -1.49(-2.72, -0.25) | 0.018 | 106.87 | <0.001 |
| Severe mental disease /psychiatric disorder | 9 | -0.78(-1.34, -0.23) | <0.001 | 62.14 | <0.001 | 0 | - | - | - | - |
| Intervention frequency | | | | | | | | | | |
| Once weekly | 21 | -0.72 (-1.04, -0.41) | <0.001 | 118.78 | <0.001 | 7 | -1.11(-1.77, -0.44) | 0.001 | 67.58 | <0.001 |
| Twice weekly | 10 | -0.79 (-1.13, -0.46) | <0.001 | 38.43 | <0.001 | 3 | -0.56(-2.49, 1.37) | 0.570 | 53.98 | <0.001 |
| ≥3 times weekly | 6 | -0.14 (-0.53, 0.25) | 0.476 | 18.65 | 0.002 | 5 | -1.67(-3.28, -0.06) | 0.042 | 185.98 | <0.001 |
| Time per session (minutes) | | | | | | | | | | |
| 15–40 | 12 | -0.52(-0.86, -0.19) | 0.002 | 59.84 | <0.001 | 9 | -1.34(-2.38, -0.29) | 0.012 | 245.42 | <0.001 |
| 41–60 | 10 | -0.56(-0.99, -0.13) | 0.012 | 62.25 | <0.001 | 6 | -0.96(-1.65, -0.27) | 0.006 | 57.46 | <0.001 |
| >60 | 12 | -0.96(-1.46, -0.47) | <0.001 | 81.18 | <0.001 | 1 | -4.1(-5.7, -2.50) | <0.001 | 0 | - |
| Country having music therapy profession | | | | | | | | | | |
| Yes | 39 | -0.65(-0.86, -0.45) | <0.001 | 234.06 | <0.001 | 13 | -1.26(-1.99, -0.53) | 0.001 | 309.93 | <0.001 |
| No | 2 | -0.83(-1.42, -0.23) | <0.001 | 0.03 | 0.864 | 3 | -1.60(-2.86, -0.34)_ | 0.003 | 16.49 | <0.001 |
| Group therapy or individual therapy | | | | | | | | | | |
| Group therapy | 30 | -0.66 (-0.92, -0.41) | <0.001 | 177.02 | <0.001 | 8 | -1.23(-2.10, -0.36) | 0.006 | 128.59 | <0.001 |
| Individual therapy | 10 | -0.67 (-1.05, -0.29) | 0.001 | 56.14 | <0.001 | 7 | -1.57(-2.71, -0.42) | 0.007 | 190.82 | <0.001 |

(*Continued*)

**Table 2.** (Continued）

| Subgroups | Music therapy | | | | | Music medicine | | | | |
|---|---|---|---|---|---|---|---|---|---|---|
| | Trials number | Effects | | Heterogeneity | | Trials number | Effects | | Heterogeneity | |
| | | SMD (95%CI) | P | I²(%) | P | | SMD (95%CI) | P | I²(%) | P |
| Setting | | | | | | | | | | |
| Outpatient | 16 | -0.89(-1.30, -0.47) | <0.001 | 103.66 | <0.001 | 12 | -1.26(-1.94, -0.57) | <0.001 | 255.53 | <0.001 |
| Inpatient | 22 | -0.57(-0.83, -0.31) | <0.001 | 127.51 | <0.001 | 3 | -0.91(-3.10, 1.28) | 0.414 | 54.87 | <0.001 |
| Used a certified music therapist | | | | | | | | | | |
| Yes | 32 | -0.69 (-0.88, -0.49) | <0.001 | 131.76 | <0.001 | - | - | - | - | - |
| No | 5 | -0.93 (-2.12, 0.25) | 0.123 | 82.69 | <0.001 | 10 | -1.71(-2.61, -0.81) | <0.001 | 234.94 | <0.001 |

remarkable different effect (Table 2). Table 2 also presents the subgroup analysis of music medicine on reducing depression.

In the subgroup analysis by total session, music therapy and music medicine both exhibited a stronger effects of short (1–4 sessions) and medium length (5–12 sessions) compared with long intervention periods (>13sessions) (Fig 6). Meta-regression demonstrated that total

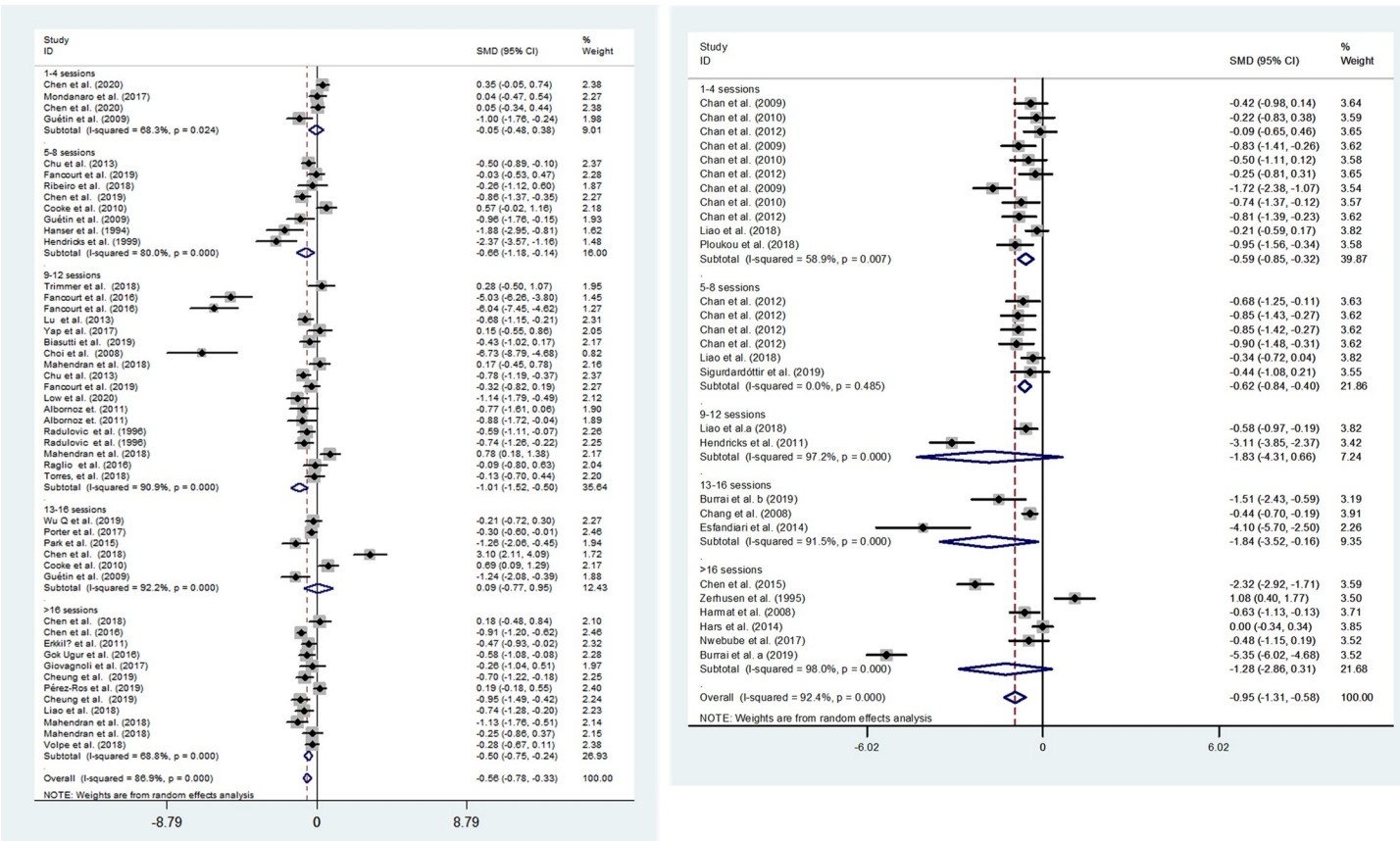

**Fig 6. Effects of music therapy and music medicine to reduce depression by total sessions.** A, evaluating the effect of music therapy; B, evaluating the effect of music medicine.

music intervention session was significantly associated with the homogeneity between studies ($P = 0.004$) (Table 3).

## Sensitivity analyses

We performed sensitivity analyses and found that re-estimating the pooled effects using fixed effect model, using trim and fill analysis, excluding the paper without information regarding music therapy, excluding the papers with more high biases, excluding the papers with small sample size ($20 < n < 30$), excluding the studies focused on the people with a severe mental disease, and excluding the papers using an infrequently used scale yielded the similar results, which indicated that the primary results was robust (Table 4).

## Evaluation of publication bias

We assessed publication bias using Egger's linear regression test and funnel plot, and the results are presented in Fig 7. For the main result, the observed asymmetry indicated that either the absence of papers with negative results or publication bias.

## Discussion

Our present meta-analysis exhibited a different effect of music therapy and music medicine on reducing depression. Different music therapy methods also exhibited a different effect, and the recreative music therapy and guided imagery and music yielded a superior effect on reducing depression compared with other music therapy methods. Furthermore, music therapy and music medicine both exhibited a stronger effects of short and medium length compared with long intervention periods. The strength of this meta-analysis was the stable and high-quality result. Firstly, the sensitivity analyses performed in this meta-analysis yielded similar results, which indicated that the primary results were robust. Secondly, considering the insufficient statistical power of small sample size, we excluded studies with a very small sample size ($n < 20$).

**Table 3. Meta-regression analysis of the main characteristics of the 33 studies.**

| Characteristics | Music therapy | | Music medicine | |
|---|---|---|---|---|
| | Coef. 95%CI | $P$ | Coef. 95%CI | $P$ |
| Sample size | 0(-0.01, 0.03) | 0.704 | 0(-0.01, 0.01) | 0.926 |
| Mean age (years) | 0.01(-0.03, 0.05) | 0.39 | - | - |
| Setting | | | | |
| Inpatient | 1 | | 1 | |
| Outpatient | 0.13(-1.98, 2.23) | 0.901 | 1.48(-0.59, 3.55) | 0.139 |
| Pre-treatment diagnosis | | | | |
| Mental health | 1 | 1 | 1 | |
| Depression | -0.24(-1.20, 0.72) | 0.622 | -0.24(-2.08, 1.61) | 0.789 |
| Severe mental disease /psychiatric disorder | -0.22(-1.18, 0.75) | 0.652 | - | |
| Music therapy method | | | | |
| Active music therapy | 1 | | | |
| Receptive music therapy | 0.13(-1.89, 2.14) | 0.895 | - | - |
| Active+receptive | 0.48(-2.26, 3.21) | 0.716 | - | - |
| Total music intervention sessions | 0.01(-0.05, 0.06) | 0.83 | -0.02(-0.03, -0.01) | 0.004 |
| Music intervention frequency | -0.08(-1.74, 1.58) | 0.918 | 0.45(-0.66, 1.57) | 0.376 |
| Time per session (minutes) | -0.01(-0.04, 0.02) | 0.482 | -0.01(-0.07, 0.05) | 0.778 |

**Table 4. Sensitivity analyses of the main outcomes [SMD (95%CI)].**

| Outcomes | Trials number | Effects | | Heterogeneity | | Egger's est | |
|---|---|---|---|---|---|---|---|
| | | SMD (95% CI) | P | $I^2(\%)$ | P | a | P |
| Music therapy | | | | | | | |
| Using fixed effect model | 41 | -0.50 (-0.58, -0.43) | <0.001 | 83 | <0.001 | -2.82(-4.71, -0.93) | 0.005 |
| Using trim and fill analysis | 41 | -0.66 (-0.86, -0.46) | <0.001 | - | <0.001 | - | - |
| Excluding the paper without information regarding music therapy (Chirico et al., 2020; Koelsch et al., 2010; Toccafondi et al., 2017; Porter et al., 2017) | 37 | -0.66 (-0.88, -0.43) | <0.001 | 82.2 | <0.001 | -3.03(-5.26, -0.81) | 0.009 |
| Excluding the papers with high bias (Toccafondi et al., 2017 and Fancourt et al., 2019) | 39 | -0.69 (-0.91, -0.47) | <0.001 | 83.6 | <0.001 | -2.95(-5.04, -0.86) | 0.007 |
| Excluding the papers with small sample size (20< n<30) | 35 | -0.57 (-0.77, -0.38) | <0.001 | 81.3 | <0.001 | 2.22(-4.53, 0.08) | 0.058 |
| Excluding the studies focused on the people with a severe mental disease (Choi et al., 2008; Cheung et al. 2019) | 32 | -0.64(-0.86, -0.42) | <0.001 | 82.1 | <0.001 | '-2.54(-4.67, -0.40) | 0.022 |
| Excluding the papers using an infrequently used scale (Erkkilä et al., 2011; Chen et al., 2015; Cheung et al., 2019; Chirico et al., 2020; Park et al., 2015; Sigurdardóttir et al., 2019; Wu et al., 2019; Low et al., 2020) | 34 | -0.62 (-0.84, -0.39) | <0.001 | 83.2 | <0.001 | -2.63(-4.67, -0.60) | 0.013 |
| Music medicine | | | | | | | |
| Using fixed effect model | 16 | -0.86(-0.98, -0.73) | <0.001 | 95.4 | <0.001 | -5.78(-11.65, 0.10) | 0.053 |
| Using trim and fill analysis | 16 | -1.33(-1.96, -0.70) | <0.001 | - | <0.001 | - | - |
| Excluding the papers with small sample size (20< n<30) [49] | 15 | -1.32(-1.98, -0.66) | <0.001 | 95.7 | <0.001 | -6.09(-12.53, 0.36) | 0.062 |
| Excluding the papers using an infrequently used scale (Chen et al., 2015) | 14 | -1.25(-1.92, -0.57) | <0.001 | 95.7 | <0.001 | -5.71(-12.38, 0.98) | 0.98 |

Some prior reviews have evaluated the effects of music therapy for reducing depression. These reviews found a significant effectiveness of music therapy on reducing depression among older adults with depressive symptoms, people with dementia, puerpera, and people with cancers [4, 5, 73–76]. However, these reviews did not differentiate music therapy from music medicine. Another paper reviewed the effectiveness of music interventions in treating depression. The authors included 26 studies and found a signifiant reduction in depression in the music intervention group compared with the control group. The authors made a clear distinction on the definition of music therapy and music medicine; however, they did not include all relevant data from the most recent trials and did not conduct a meta-analysis [77]. A recent meta-analysis compared the effects of music therapy and music medicine for reducing depression in people with cancer with seven RCTs; the authors found a moderately strong, positive impact of music intervention on depression, but found no difference between music therapy and music medicine [78]. However, our present meta-analysis exhibited a different effect of music therapy and music medicine on reducing depression, and the music medicine yielded a superior effect on reducing depression compared with music therapy. The different effect of music therapy and music medicine might be explained by the different participators, and nine studies used music therapy to reduce the depression among people with severe mental disease /psychiatric disorder, while no study used music medicine. Furthermore, the studies evaluating music therapy used more clinical diagnostic scale for depressive symptoms.

A meta-analysis by Li et al. [74] suggested that medium-term music therapy (6–12 weeks) was significantly associated with improved depression in people with dementia, but not short-

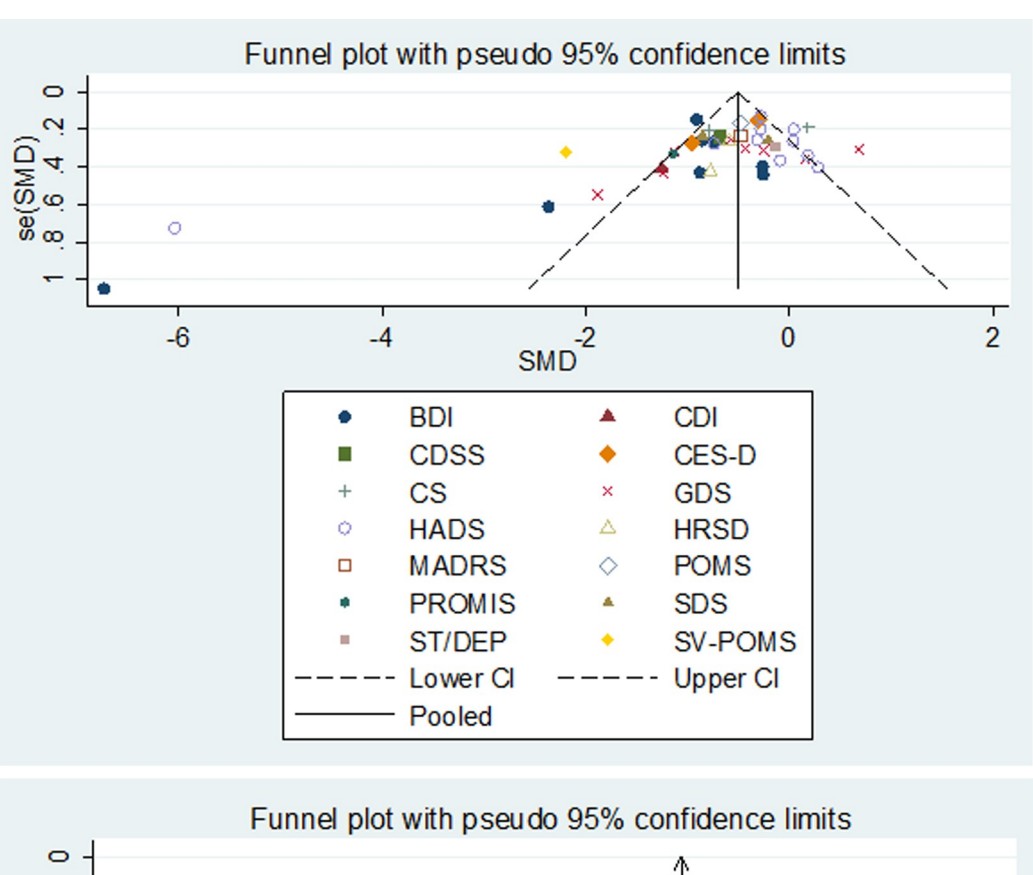

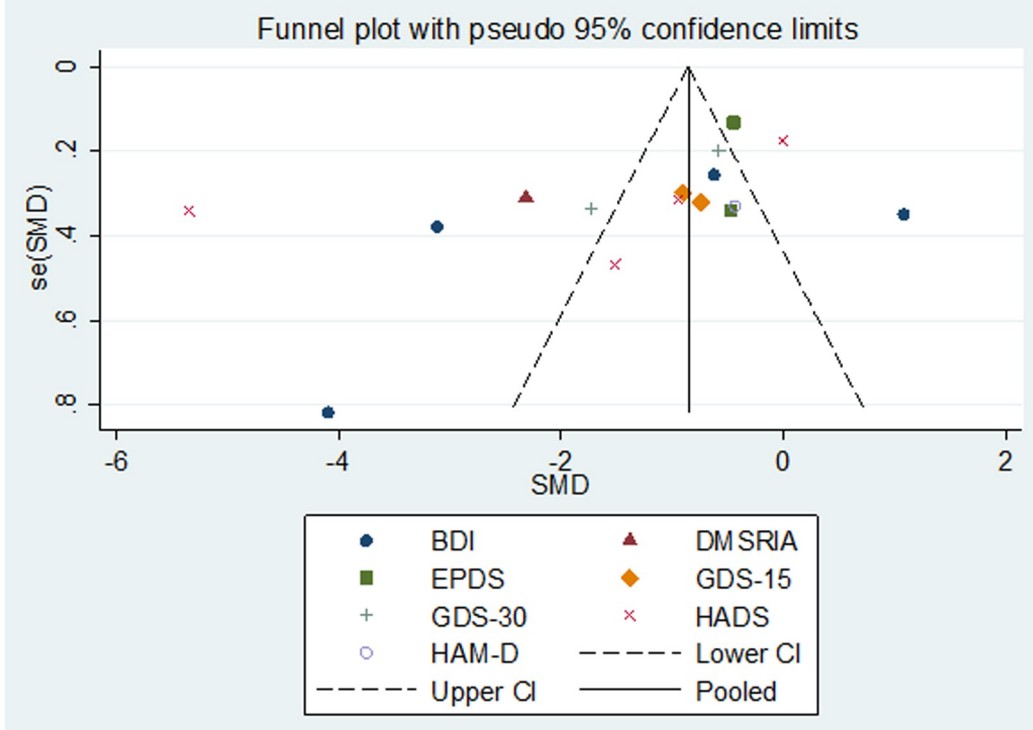

**Fig 7. Funnel plot illustrating proneness to publication bias for the included studies.** A, evaluating the publication bias of music therapy; B, evaluating the publication bias of music medicine; BDI = Beck Depression Inventory; CDI = Children's Depression Inventory; CDSS = depression scale for schizophrenia; CES-D = Center for Epidemiologic Studies Depression; CS = Cornell Scale; DMSRIA = Depression Mood Self-Report Inventory for Adolescence; EPDS = Edinburgh Postnatal Depression Scale; GDS-15 = Geriatric Depression Scale-15; GDS-30 = Geriatric Depression Scale-30; HADS = Hospital Anxiety and Depression Scale; HRSD (HAMD) = Hamilton Rating Scale for Depression; MADRS = Montgomery-sberg

Depression Rating Scale; PROMIS = Patient Reported Outcomes Measurement Information System; SDS = Self-Rating Depression Scale; State-Trait Depression Questionnaire = ST/DEP; SV-POMS = short version of Profile of Mood Stat.

term music therapy (3 or 4 weeks). On the contrary, our present meta-analysis found a stronger effect of short-term (1–4 weeks) and medium-term (5–12 weeks) music therapy on reducing depression compared with long-term (≥13 weeks) music therapy. Consistent with the prior meta-analysis by Li et al., no significant effect on depression was observed for the follow-up of one or three months after music therapy was completed in our present meta-analysis. Only five studies analyzed the therapeutic effect for the follow-up periods after music therapy intervention therapy was completed, and the rather limited sample size may have resulted in this insignificant difference. Therefore, whether the therapeutic effect was maintained in reducing depression when music therapy was discontinued should be explored in further studies. In our present meta-analysis, meta-regression results demonstrated that no variables (including period, frequency, method, populations, and so on) were significantly associated with the effect of music therapy. Because meta-regression does not provide sufficient statistical power to detect small associations, the non-significant results do not completely exclude the potential effects of the analyzed variables. Therefore, meta-regression results should be interpreted with caution.

Our meta-analysis has limitations. First, the included studies rarely used masked methodology due to the nature of music therapy, therefore the performance bias and the detection bias was common in music intervention study. Second, a total of 13 different scales were used to evaluate the depression level of the participators, which may account for the high heterogeneity among the trials. Third, more than half of those included studies had small sample sizes (<50), therefore the result should be explicated with caution.

## Conclusion

Our present meta-analysis of 55 RCTs revealed a different effect of music therapy and music medicine, and different music therapy methods also exhibited a different effect. The results of subgroup analyses revealed that the characters of music therapy were associated with the therapeutic effect, for example specific music therapy methods, short and medium-term therapy, and therapy with more time per session may yield stronger therapeutic effect. Therefore, our present meta-analysis could provide suggestion for clinicians and policymakers to design therapeutic schedule of appropriate lengths to reduce depression.

## Supporting information

**S1 Checklist. PRISMA checklist.**
(DOC)

**S1 Dataset.**
(XLSX)

## Author Contributions

**Conceptualization:** Qishou Tang, Peijie Ye.

**Methodology:** Zhaohui Huang.

**Software:** Zhaohui Huang.

**Writing – original draft:** Qishou Tang.

**Writing – review & editing:** Huan Zhou, Peijie Ye.

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
