## [Decision Letter · Decision Letter 0]

5 Aug 2020

PONE-D-20-17706

Effects of music therapy on depression: a meta-analysis of randomized controlled trials

PLOS ONE

Dear Dr. Ye,

Thank you for submitting your manuscript to PLOS ONE. After careful consideration, we feel that it has merit but does not fully meet PLOS ONE’s publication criteria as it currently stands. Therefore, we invite you to submit a revised version of the manuscript that addresses the points raised during the review process.

We look forward to receiving your revised manuscript.

Kind regards,

Sukru Torun

Academic Editor

PLOS ONE

Additional Editor Comments:

Dear Author,

Thank you for your valuable submission. I think it would be appropriate to emphasize the main problem first. Various musical interventions are used in medical settings to improve the patient's well-being, and of course, there are many publications on this subject. However, it is important to properly differentiate between these interventions for some important reasons I have pointed out below.

The music therapy definition you made, as "Music therapy was defined as music therapy provided by a qualified music teacher, psychological therapist, or nurse" is not universally accepted specific definition for music therapy. Moreover, the specific methods used in receptive music therapy include music-assisted relaxation, music and imagery, and Guided Imagery and Music (Bonny Method). Each of these may have different levels of effects on depression. It is not clear that which receptive music therapy studies in your review have used which of these methods. So, the majority of studies that you accepted as the receptive music therapy seems to be music medicine studies indeed. Similar critiques may also be apply to some of the studies you describe as active music therapy. Today, it is widely accepted that these music-based interventions should be divided into two major categories, namely music therapy (MT) and music medicine (MM). MM mainly based on patients' pre-recorded or rarely listening to live music and the direct effects of the music they listen to. In other words, MM aims to use music like medicines. It often managed by a medical professional other than a music therapist, and not needed a therapeutic relationship with the patients. Conversely, music therapy is the clinical and evidence-based use of music interventions to accomplish individualized goals within a therapeutic relationship by a credentialed music therapist who has completed an approved music therapy program. So, music therapy is a relational, interaction based form of therapy within a therapeutic relationship between the therapist and the client, and includes the triad of the music, the client and the music therapist. Since music therapy interventions is an evidence-based procedure using special music therapy methods of interventions and a more pragmatic approach than other music-based interventions, their effect levels and results are also different.

In the context of the above mentioned explanations, it is clear that to evaluate the effects of music therapy and other music based intervention studies together on depression can be misleading. The subjects I have mentioned so far have never been addressed in the introduction and discussion sections of your manuscript. I think that will be perceived as a major deficiency at least by the readers who are closer to the subject. In this sense, I think that an attentive revision considering the following views will be valuable and needed:

- The universally accepted definitions of music therapy (including active and receptive music therapy) and music medicine should be taken into account.

- It should be clarified that how many studies in your review did included a certified music therapist.

- Analyses, results and discussion should be submitted to the readers in accordance with all this distinctions and definitions. (The way to this seems to be to compare the effects of music medicine and music therapy on depression in parallel with the possible differences of music interventions used, and to discuss their possible implications on the results.)

- Another important point is that you did not mention nor discuss any of important reviews on same subject (for example please see: https://www.cochranelibrary.com/cdsr/doi/10.1002/14651858.CD004517.pub3/epdf/full or https://www.frontiersin.org/articles/10.3389/fpsyg.2017.01109/full or https://www.cochranelibrary.com/cdsr/doi/10.1002/14651858.CD006911.pub3/full)

I am aware that such a major revision will, in a sense, be a challenging way that may require a new analysis of your data. However, I believe you would appreciate that a study aimed at shedding light on potential music-based interventions in an important public health problem such as depression should not be misleading.

Thank you for your effort in advance.

Annotate:

Besides, according to the statistical reviewer who only reviewed the statistical approach used in this paper, there are two caveats:

1. The authors state that they excluded studies with fewer than 20 participants in one place in the paper (page 4), but fewer than 30 participants in another place in the paper (Table 4). This needs to be corrected for consistency.

2. The authors mention stronger effects of short and medium length vs. long music therapy periods in their results but there is no accompanying figure. I think it would be beneficial to show these findings in a figure (Forest plot).

Journal Requirements:

"This work was supported by the Key Project of University Humanities and Social Science

Research in Anhui Province (SK2017A0191), Research Project of Anhui Province Social Science

Innovation Development (2018XF155), Ministry of Education Humanities and Social Sciences

Research Youth fund Project (17YJC840033)."

5. Please amend your list of authors on the manuscript to ensure that each author is linked to an affiliation. Authors’ affiliations should reflect the institution where the work was done (if authors moved subsequently, you can also list the new affiliation stating “current affiliation:….” as necessary).

6. Please ensure that you refer to Figure 5 in your text as, if accepted, production will need this reference to link the reader to the figure.

Reviewers' comments:

Reviewer's Responses to Questions

**Comments to the Author**

1. Is the manuscript technically sound, and do the data support the conclusions?

Reviewer #1: Yes

Reviewer #2: Partly

Reviewer #3: Yes

2. Has the statistical analysis been performed appropriately and rigorously? 

Reviewer #1: Yes

Reviewer #2: No

Reviewer #3: Yes

3. Have the authors made all data underlying the findings in their manuscript fully available?

Reviewer #1: Yes

Reviewer #2: Yes

Reviewer #3: Yes

4. Is the manuscript presented in an intelligible fashion and written in standard English?

Reviewer #1: Yes

Reviewer #2: No

Reviewer #3: Yes

5. Review Comments to the Author

Reviewer #1: Thank you for conducting this research and submitting it for publication consideration.

I recognize that English may not be the primary language of the authors. There are a few instances where the language could be improved, but that is mostly a copy-editing issue. There is also a lot of passive voice in the paper. I recommend making the voice active. This will enhance the readability of the paper.

I have a few comments that I hope will improve the paper.

1. Not all countries have an established music therapy profession. I recognize that this creates challenges for the authors! I'm wondering if the authors might consider including this as a factor in the analysis? For example, if a nurse provides "music therapy" in a country that does not have music therapy as a profession, is the effect equivalent as when a qualified music therapist in a country that has music therapy as a profession provides it? This might provide some incentive for occupational regulation and establishing professional music therapy associations.

2. please fix the "short title" (oxygen)

3. Music therapy with fewer minutes might yield superior effects. This may be misleading. Is there a minimum number of minutes? How many minutes might be optimal for therapeutic outcome? I believe it does make sense that longer sessions may result in less impact - quantity/duration does not always result in enhanced outcome.

4. I believe a stronger case needs to be made for the study. There are existing meta-analyses of MT for depression (Aalbers et al., 2017 Cochrane Review). What makes the current study unique and different? What are the gaps in the literature that warrant this study? Have there been a lot of recent additions to the literature that warrant a new meta-analysis?

5. A stronger discussion of the limitation of this study. Many studies did not evaluate a group with major depression/major depressive disorder (music therapy for chronic pain is important, but the variance of the populations under study does constitute a limitation). So, this study is not exclusive to adults with a major mental health condition. Might effects be different for people who are depressed versus people who are not depressed?

6. Instead of "blinding/blinded" please use "masking/masked."

7. Is there a citation that supports your classification of active versus receptive? (I would think Bruscia would be a good place to start with that...)

8. One item that I am not seeing is group therapy versus individual therapy. Did the authors screen for that? If so, is there an optimal group size? Are effects stronger when in a group format versus an individual format? This would have serious implications for clinical practice.

9. What about inpatient settings (such as a secure [locked] unit at a mental health facility) versus outpatient settings?

10. One item that I believe is missing is the dose. Not necessarily the duration (number of minutes) of each session, but the total number of sessions a participant has received. Gold has done some work in this area. Is there is a certain number of sessions that are needed to reach a therapeutic outcome? The number of sessions/week is good, but the number of total sessions is important.

11. Table 1 has the mean age. I recommend including the SD as well.

Thank you for taking the time to consider these suggestions. While receiving critical feedback can be difficult, please understand that my intentions are to improve the paper and ensure it has maximum impact. This is an important addition to the literature and I am grateful to the authors for their scholarship. I wish you the best!

Reviewer #2: This article addresses an important topic that is of interest to music therapists, psychiatrists and teachers and metal health practitioners. The statistics look promising. However, the major concern is that the definition of music therapy is theoretically and practically incorrect and misleading:

"7 Music therapy was defined as music therapy provided by a qualified music teacher, psychological

8 therapist, or nurse. " The study is missing several research studies that I am aware of and this makes its content suspicious. Also missing is a more depth-ful analysis of what active and passive music therapy is, and if it is indeed performed by those in other professions who have no training in 'musuc therapy;'-than the contents and findings are misleading and irrelevant.

Reviewer #3: I only reviewed the statistical approach used in this paper, which appeared appropriate for the research question under study. There are two caveats:

1. The authors state that they excluded studies with fewer than 20 participants in one place in the paper (page 4), but fewer than 30 participants in another place in the paper (Table 4). This needs to be corrected for consistency.

2. The authors mention stronger effects of short and medium length vs. long music therapy periods in their results but there is no accompanying figure. I think it would be beneficial to show these findings in a figure (Forest plot).

6. PLOS authors have the option to publish the peer review history of their article (what does this mean?). If published, this will include your full peer review and any attached files.

Reviewer #1: No

Reviewer #2: No

Reviewer #3: No

---

## [Author Response · Author response to Decision Letter 0]

29 Sep 2020

Response to Reviewers

Dear Editors and Reviewers:

Thank you for your letter and for the reviewers’ comments concerning our manuscript entitled " Effects of music therapy on depression: a meta-analysis of randomized controlled trials (PONE-D-20-17706)". 

Those comments are all valuable and very helpful for revising and improving our paper, as well as the important guiding significance to our researches. We have studied comments carefully and have made revision which we hope meet with approval. All the revised portions were marked in red font in the new document. The main corrections in the paper and the responds to the reviewer’s comments are as flowing:

Additional Editor Comments:

Dear Author,

Thank you for your valuable submission. I think it would be appropriate to emphasize the main problem first. Various musical interventions are used in medical settings to improve the patient's well-being, and of course, there are many publications on this subject. However, it is important to properly differentiate between these interventions for some important reasons I have pointed out below.

 The music therapy definition you made, as "Music therapy was defined as music therapy provided by a qualified music teacher, psychological therapist, or nurse" is not universally accepted specific definition for music therapy. Moreover, the specific methods used in receptive music therapy include music-assisted relaxation, music and imagery, and Guided Imagery and Music (Bonny Method). Each of these may have different levels of effects on depression. It is not clear that which receptive music therapy studies in your review have used which of these methods. So, the majority of studies that you accepted as the receptive music therapy seems to be music medicine studies indeed. Similar critiques may also be apply to some of the studies you describe as active music therapy. Today, it is widely accepted that these music-based interventions should be divided into two major categories, namely music therapy (MT) and music medicine (MM). MM mainly based on patients' pre-recorded or rarely listening to live music and the direct effects of the music they listen to. In other words, MM aims to use music like medicines. It often managed by a medical professional other than a music therapist, and not needed a therapeutic relationship with the patients. Conversely, music therapy is the clinical and evidence-based use of music interventions to accomplish individualized goals within a therapeutic relationship by a credentialed music therapist who has completed an approved music therapy program. So, music therapy is a relational, interaction based form of therapy within a therapeutic relationship between the therapist and the client, and includes the triad of the music, the client and the music therapist. Since music therapy interventions is an evidence-based procedure using special music therapy methods of interventions and a more pragmatic approach than other music-based interventions, their effect levels and results are also different.

 In the context of the above mentioned explanations, it is clear that to evaluate the effects of music therapy and other music based intervention studies together on depression can be misleading. The subjects I have mentioned so far have never been addressed in the introduction and discussion sections of your manuscript. I think that will be perceived as a major deficiency at least by the readers who are closer to the subject. In this sense, I think that an attentive revision considering the following views will be valuable and needed: 

Response：We have studied comments carefully and revised the manuscript extensively according to the reviewer’s comments.

Firstly, We have amended the music therapy definition mainly based on the World Federation of Music Therapy (WFMT) and The American Music Therapy Association (AMTA), WFMT defines music therapy as “the professional use of music and its elements as an intervention inmedical, educational, and everyday environments with individuals, groups, families, or communities who seek to optimize their quality of life and improve their physical, social,communicative, emotional, intellectual, and spiritual health and wellbeing”. AMTA defines music therapy as “Music Therapy is the clinical and evidence-based use of music interventions to accomplish individualized goals within a therapeutic relationship by a credentialed professional who has completed an approved music therapy program”. [American Music Therapy Association (2020). Definition and Quotes about Music Therapy. Available online at: https://www.musictherapy.org/about/quotes/ (Accessed Sep 13, 2020).][van der Steen, J. T., et al. (2017). "Music-based therapeutic interventions for people with dementia." Cochrane Database Syst Rev 5: CD003477.]

Secondly, we have re-studed all included papers carefully and added the specific intervention methods of each paper in table 1 (Table 1. Characteristics of clinical trials included in this meta-analysis). Two main types of music therapy were distinguished in our present study - receptive (or passive) and active music therapy. The specific methods used in receptive music therapy in our included papers including music-assisted relaxation, music and imagery, and guided imagery and music (Bonny Method), while the specific methods used in active music therapy included recreative music therapy, improvisational music therapy, song writing, and so on.

Thirdly, we have added some contents regarding the distinction between music therapy and music medicine in introduction and discussion sections of our manuscript. 

The following contents are added in introduction section, “Today, it is widely accepted that the music-based interventions should be divided into two major categories, namely music therapy and music medicine. According to the American Music Therapy Association (AMTA), “music therapy is the clinical and evidence-based use of music interventions to accomplish individualized goals within a therapeutic relationship by a credentialed professional who has completed an approved music therapy program”. Therefore, music therapy is an established health profession in which music is used within a therapeutic relationship to address physical, emotional, cognitive, and social needs of individualst, and includes the triad of the music, the client and the qualified music therapist. [American Music Therapy Association (2020). Definition and Quotes about Music Therapy. Available online at: https://www.musictherapy.org/about/quotes/ (Accessed Sep 13, 2020).] While, music medicine is defined as mainly listening to prerecorded music provided by medical personnel or rarely listening to live music. In other words, music medicine aims to use music like medicines. It often managed by a medical professional other than a music therapist, and not needed a therapeutic relationship with the patients. Therefore, the essential difference of music therapy and music medicine is whether a therapeutic relationship is developed between a trained music therapist and the client. 

[Bradt, J., et al. (2015). "The impact of music therapy versus music medicine on psychological outcomes and pain in cancer patients: a mixed methods study." Supportive care in cancer : official journal of the Multinational Association of Supportive Care in Cancer 23(5): 1261-1271. 

[Yinger, O. S. and L. Gooding (2014). "Music therapy and music medicine for children and adolescents." Child and adolescent psychiatric clinics of North America 23(3): 535-553.] 

【Tony Wigram．Inge Nyggard Pedersen＆Lars Ole Bonde，A Compmhensire Guide to Music Therapy．London and Philadelphia：Jessica Kingsley Publishen．2002：143．】

In the context of the clear distinction between these two major cagerories, it is clear that to evaluate the effects of music therapy and other music based intervention studies together on depression can be misleading. While, the distinction was not always clear in most of prior papers, and we found that no meta-analysis comparing the effects of music therapy and music medicine was conducted. Just a few studies made a comparison of music-based interventions on psychological outcomes between music therapy and music medicine. We aimed to (1) compare the effect between music therapy and music medicine on depression; (2) compare the effect between different specific methods used inmusic therapy on depression; (3) compare the effect of music-based interventions on depression among different population. 

 [Bradt, J., et al. (2015). "The impact of music therapy versus music medicine on psychological outcomes and pain in cancer patients: a mixed methods study." Supportive care in cancer : official journal of the Multinational Association of Supportive Care in Cancer 23(5): 1261-1271.[Yinger, O. S. and L. Gooding (2014). "Music therapy and music medicine for children and adolescents." Child and adolescent psychiatric clinics of North America 23(3): 535-553.] 

The last, we have made a new analysis of our data. 1) including three new papers and re-analying of our data, 2) adding the comparison of music therapy and music medicine, 3) adding the comparison of impatient setting and outpatients setting, 4) adding the comparison of depressed people and not depressed people, 5）adding the comparison of countries have having music therapy profession and not, 6) adding the comparison of group therapy and individual therapy, 7) added the comparison of different intervention dose, and so on. 

 - The universally accepted definitions of music therapy (including active and receptive music therapy) and music medicine should be taken into account.

Response: (1)We have amended the of definitions of music therapy. The revised difinitons of music therapy was “Music Therapy is the clinical and evidence-based use of music interventions to accomplish individualized goals within a therapeutic relationship by a credentialed professional who has completed an approved music therapy program”. [American Music Therapy Association (2020). Definition and Quotes about Music Therapy. Available online at: https://www.musictherapy.org/about/quotes/ (Accessed Sep 13, 2020).]

We have added some contents on the distinction between music therapy (MT) and music medicine (MM) in introduction and discussion sections of our manuscript. 

We have added the analysis of the comparion of music therapy (MT) and music medicine (MM) in Methord and Results sections 

- It should be clarified that how many studies in your review did included a certified music therapist.

Response: we have re-studed all included papers carefully and added a new varible (Intervenor or therapist) into table 1, and the corresponding description was addded in the results section. Of 55 studies, 32 used a certified music therapist, 15 not used a certified music therapist (for example researcher, nurse), and 10 not reported relevent information. 

- Analyses, results and discussion should be submitted to the readers in accordance with all this distinctions and definitions. (The way to this seems to be to compare the effects of music medicine and music therapy on depression in parallel with the possible differences of music interventions used, and to discuss their possible implications on the results.)

Response: We have divided music-based interventions into two major categories, namely music therapy and music medicine according to the difinition. With respect to specific methods used in music therapy, we also have divided music therapy into receptive (or passive) and active music therapy. The specific methods used in receptive music therapy in our included papers including music-assisted relaxation, music and imagery, and guided imagery and music (Bonny Method), and the specific methods used in active music therapy included recreative music therapy and improvisational music therapy.

We have added some sub-group analyses by different music intervention categories, different music therapy categories, and specific music therapy methords.

The the above mentioned content have been added to Intruduction Analyses, results and discussion section. 

- Another important point is that you did not mention nor discuss any of important reviews on same subject (for example please see: https://www.cochranelibrary.com/cdsr/doi/10.1002/14651858.CD004517.pub3/epdf/full or https://www.frontiersin.org/articles/10.3389/fpsyg.2017.01109/full or https://www.cochranelibrary.com/cdsr/doi/10.1002/14651858.CD006911.pub3/full)

Response: we are very sorry for not mentioning these important reviews. We have studied these reviews carefully and discussed these reviews in Discussion sections. 

Some prior reviews have evaluated the effects of music therapy for reducing depression. Aalbers and colleagues included nine studies in their review; they concluded that music therapy provides short-term benefificial effects for people with depression, and suggested that high-quality trials with large sample size were needed. However, this review was limited to studies of individuals with a diagnosis of depression, and did not differentiate music therapy from music medicine. Another paper reviewed the effectiveness of music interventions in treating depression. The authors included 26 studies and found a signifiant reduction in depression in the music intervention group compared with the controp group. The authors made a clear distincition on the definition of music therapy and music medicine; however, they did not include all relevant data from the most recent trials and did not conduct a meta-analysis. A recent meta-analysis compared the effects of music therapy and music medicine for reducing depression in people with cancer with seven RCTs; the authors found a moderately strong, positive impact of music intervention on depression , but found no difference between music therapy and music medicine. 

【Aalbers, S., et al. (2017). "Music therapy for depression." Cochrane Database Syst Rev 11: CD004517.】

【Leubner, D. and T. Hinterberger (2017). "Reviewing the Effectiveness of Music Interventions in Treating Depression." Front Psychol 8: 1109.】

【Bradt, J., et al. (2016). "Music interventions for improving psychological and physical outcomes in cancer patients." Cochrane Database Syst Rev(8): CD006911.】

To date, many new trials focued on music therapy and depression in differnt poupulation (such as people with cancer, people with dementia, people with chronic disease, and so on ) have been performed, but they have not yet been systematically reviewed. 

I am aware that such a major revision will, in a sense, be a challenging way that may require a new analysis of your data. However, I believe you would appreciate that a study aimed at shedding light on potential music-based interventions in an important public health problem such as depression should not be misleading.

Thank you for your effort in advance.

Response: Those comments are all valuable and very helpful for revising and improving our paper, as well as the important guiding significance to our researches. We have studied comments carefully and have made revision according to the comments. 

Annotate:

Besides, according to the statistical reviewer who only reviewed the statistical approach used in this paper, there are two caveats:

1. The authors state that they excluded studies with fewer than 20 participants in one place in the paper (page 4), but fewer than 30 participants in another place in the paper (Table 4). This needs to be corrected for consistency.

Response: We are sorry for making this mistake. In the Methord section, we defined exclusive criteria as studies with a very small sample size (n<20),while in table 4 we performed the sensitivity analyses by excluding the papers with smale sample size ( 20< n<30). We have amended the table 4. 

2. The authors mention stronger effects of short and medium length vs. long music therapy periods in their results but there is no accompanying figure. I think it would be beneficial to show these findings in a figure (Forest plot).

Response: We have added these findings with a forest plot (figure 6) according to the comment. 

Journal Requirements:

Response: We have amended our manuscript according to PLOS ONE's style requirements

Please include your tables as part of your main manuscript and remove the individual files. Please note that supplementary tables (should remain/ be uploaded) as separate "supporting information" files.

Response: We have adjusted these content according to the comment. 

"This work was supported by the Key Project of University Humanities and Social Science

Research in Anhui Province (SK2017A0191), Research Project of Anhui Province Social Science

Innovation Development (2018XF155), Ministry of Education Humanities and Social Sciences

Research Youth fund Project (17YJC840033)."

Response: We would like to update our funding statement as follows: The funders had a role in study design, text editing, interpretation of results, decision to publish and preparation of the manuscript. 

4.LOS requires an ORCID iD for the corresponding author in Editorial Manager on papers submitted after December 6th, 2016. Please ensure that you have an ORCID iD and that it is validated in Editorial Manager. To do this, go to ‘Update my Information’ (in the upper left-hand corner of the main menu), and click on the Fetch/Validate link next to the ORCID field. This will take you to the ORCID site and allow you to create a new iD or authenticate a pre-existing iD in Editorial Manager. Please see the following video for instructions on linking an ORCID iD to your Editorial Manager account: https://www.youtube.com/watch?v=_xcclfuvtxQ

Response: We have created a new ORCID iD accordingly to your instructions. 

5. Please amend your list of authors on the manuscript to ensure that each author is linked to an affiliation. Authors’ affiliations should reflect the institution where the work was done (if authors moved subsequently, you can also list the new affiliation stating “current affiliation:….” as necessary).

Response: we are sorry for making this mistake, we have amended our list of authors on the manuscript accordingly.

6. Please ensure that you refer to Figure 5 in your text as, if accepted, production will need this reference to link the reader to the figure.

Response: We have checked the refer to Figure 5 and found that the refer to figure 5 was a mistake, and we have amended it. 

 Response: we only have a Supporting Information files (PRISMA-2009-Checklist), and we have added the captions for this Supporting Information files accordingly. We also have updated in-text citations to match accordingly.

Reviewers' comments:

Reviewer's Responses to Questions

Comments to the Author

1. Is the manuscript technically sound, and do the data support the conclusions?

Reviewer #1: Yes

Reviewer #2: Partly

Reviewer #3: Yes

2. Has the statistical analysis been performed appropriately and rigorously?

Reviewer #1: Yes

Reviewer #2: No

Reviewer #3: Yes

3. Have the authors made all data underlying the findings in their manuscript fully available?

Reviewer #1: Yes

Reviewer #2: Yes

Reviewer #3: Yes

4. Is the manuscript presented in an intelligible fashion and written in standard English?

Reviewer #1: Yes

Reviewer #2: No

Reviewer #3: Yes

5. Review Comments to the Author

Reviewer #1: Thank you for conducting this research and submitting it for publication consideration.

Response: Thinks very much for your comment.

I recognize that English may not be the primary language of the authors. There are a few instances where the language could be improved, but that is mostly a copy-editing issue. There is also a lot of passive voice in the paper. I recommend making the voice active. This will enhance the readability of the paper.

Response: Thinks very much for your comment. Our manuscript have been edited for proper English language, grammar, punctuation, spelling, and overall style by one qualified native English speaking editors.

I have a few comments that I hope will improve the paper.

1. Not all countries have an established music therapy profession. I recognize that this creates challenges for the authors! I'm wondering if the authors might consider including this as a factor in the analysis? For example, if a nurse provides "music therapy" in a country that does not have music therapy as a profession, is the effect equivalent as when a qualified music therapist in a country that has music therapy as a profession provides it? This might provide some incentive for occupational regulation and establishing professional music therapy associations. 

Response: This suggestion is valuable and we have tried to judge if the countries in our inluded papers have an established music therapy profession by checking the author's work address, literature review, visiting the important website about music therapy, and consulting to some famous music therapist via emails. The following table showed that four countries may be not have a music therapy profession. We have added the comparison of the country having music therapy profession and not. 

https://erikdalton.com/find-a-certified-therapist/

https://www.musictherapy.org/about/listserv/

Table 1. The information on the music therapy profession in the inluded papers

Country Country having music therapy profession

Korea Korean Music Therapy Association

South Korea Korean Music Therapy Association

UK British Association for Music Therapy

Australia Australian Music Therapy Association

Canada Canadian Association of Music Therapists

China Chinese Professional Music Therapist Association

Taiwan China Chinese Professional Music Therapist Association

Denmark Dansk forbund for musikterapie

Finland Finnish Society for Music Therapy

Hong Kong China Hong Kong Music Therapy and Counseling Association 

Serbia Music Therapists of Serbia organize workshops

Switzerland Swiss Association of Music Therapy

USA The American Music Therapy Association

Singapore The Association for Music Therapy (Singapore)

Brazil Uniao Braileira Das Associacoes De Musicoterapia

France YES

Germany YES

Italy YES

Northern Ireland YES

Spain YES

Spanish YES

Turkey YES

Greece No

Hungary No

Iran No

Venezuela No

2.please fix the "short title" (oxygen)

Response: We’re sorry for making this mistake, and we have corrected this mistake. 

Music therapy with fewer minutes might yield superior effects. This may be misleading. Is there a minimum number of minutes? How many minutes might be optimal for therapeutic outcome? I believe it does make sense that longer sessions may result in less impact - quantity/duration does not always result in enhanced outcome.

Response: In 33 included trials, intervention time each session was different, the mimimum time was 15 minutes in only one study (Burrai et al., 2019b), followed by 20 minuters in four studies (Chirico et al., 2020; Guétin et al., 2009; Hanser et al., 1994; Sigurdardóttir et al., 2019). In our subgroup analysis by time per session (minutes), we divided time per session into three groups, namely 15-40, 41-60, >60, and this presentation might be unclear. 

In order to respond this comment, we have re-divided the time per session into four groups, namely 15-40, 41-60, 61-120, to explore the optimal minuter per session for therapeutic outcome.

I believe a stronger case needs to be made for the study. There are existing meta-analyses of MT for depression (Aalbers et al., 2017 Cochrane Review). What makes the current study unique and different? What are the gaps in the literature that warrant this study? Have there been a lot of recent additions to the literature that warrant a new meta-analysis?

Response: Some prior reviews have evaluated the effects of music therapy for reducing depression. Aalbers and colleagues (Aalbers et al., 2017）included nine studies in their review; they concluded that music therapy provides short-term benefificial effects for people with depression, and suggested that high-quality trials with large sample size were needed. However, this review was limited to studies of individuals with a diagnosis of depression, and did not differentiate music therapy from music medicine. 

Another paper reviewed the effectiveness of music interventions in treating depression. The authors (Leubner D., 2017) included 26 studies and found a signifiant reduction in depression in the music intervention group compared with the controp group. The authors made a clear distincition on the definition of music therapy and music medicine; however, they did not include all relevant data from the most recent trials and did not conduct a meta-analysis. A recent meta-analysis (Bradt et al., 2016) compared the effects of music therapy and music medicine for reducing depression with seven RCTs; the authors found a moderately strong, positive impact of music intervention on depression , but found no difference between music therapy and music medicine. However, this review was limited to studies of individuals with a diagnosis of cancer.

【Aalbers, S., et al. (2017). "Music therapy for depression." Cochrane Database Syst Rev 11: CD004517.】

【Leubner, D. and T. Hinterberger (2017). "Reviewing the Effectiveness of Music Interventions in Treating Depression." Front Psychol 8: 1109.】

【Bradt, J., et al. (2016). "Music interventions for improving psychological and physical outcomes in cancer patients." Cochrane Database Syst Rev(8): CD006911.】

Figure 1 presents the number of published paper ( search from Pubmed) focued on music therapy and depression from 1983 to 2020, the published paper was in the rapidly growing stage during the past five years. While, the above mentioned reviews all included papers published before 2017. To date, many new trials focued on music therapy and depression in differnt poupulation (such as people with cancer, people with dementia, people with chronic disease, and so on ) have been performed, but they have not yet been systematically reviewed. 

While, no meta-analysis compared the the difference of music therapy on depression in differnt poupulation (such as people with depression, people with dementia, people with chronic disease, health people, and so on ) have been performed. 

Figure 1 The pubished papers from 1983 to 2020 focused on music therapy and depression (searched from Pubmed)

In our persent meta-analysis, we aimed to (1) compare the effect between music therapy and music medicine on depression; (2) compare the effect between different specific methods used inmusic therapy on depression; (3) compare the effect of music-based interventions on depression among different population. 

We have added the above content to Intruduction and Dissussion sections. 

5.A stronger discussion of the limitation of this study. Many studies did not evaluate a group with major depression/major depressive disorder (music therapy for chronic pain is important, but the variance of the populations under study does constitute a limitation). So, this study is not exclusive to adults with a major mental health condition. Might effects be different for people who are depressed versus people who are not depressed?

Response: This is a very important comment. According to this comment, we have made some revision. 

Firstly, we have added a sensitivity analysis by excluding the studes focused on the people with a major mental health condition. 

Secondly, we have re-grouped the populations into three groups, namely mental health, severe mental disease /psychiatric disorder, and depression and we have added the subgroup analysis (table 2 in revised manuscript).. 

Thirdly, we have added the analysis of the difference between people who are depressed versus people who are not depressed accordingly (table 2 in revised manuscript). 

6.Instead of "blinding/blinded" please use "masking/masked."

Response: We have replaced "blinding/blinded" with "masking/masked" according to this comment. 

7. Is there a citation that supports your classification of active versus receptive? (I would think Bruscia would be a good place to start with that...)

Response: In active methods (improvisational, re-creative, compositional), participants are ‘making music’ , and in receptive music therapy (music-assisted relaxation, music and imagery, guided imagery and music, lyrics analysis ), participants are ‘receiving’ (e.g. listening to) music (Bruscia 2014; Wheeler 2015).

We have amended the difinition and added the citation to the Result section according to this commment. 

[Bruscia KE. Defining Music Therapy. 3rd Edition.University Park, Illinois, USA: Barcelona Publishers, 2014.]

[Wheeler BL. Music Therapy Handbook. New York, New York, USA: Guilford Publications, 2015.]

8. One item that I am not seeing is group therapy versus individual therapy. Did the authors screen for that? If so, is there an optimal group size? Are effects stronger when in a group format versus an individual format? This would have serious implications for clinical practice.

Response: Of the 55 studies, 38 used group therapy, 17 used individual therapy, and 2 not reported. We have added the comparison of group therapy versus individual therapy according to this comment (table 2 in revised manuscript). 

9. What about inpatient settings (such as a secure [locked] unit at a mental health facility) versus outpatient settings?

Response: Of 55 studies, a total of 25 studies were conducted in impatient setting,28 studies were in outpatients setting setting, and 2 studies not repoted the setting. We have added the subgroup analysis by inpatient settings (secure [locked] unit at a mental health facility versus outpatient settings) according to this comment (table 2 in revised manuscript). 

10. One item that I believe is missing is the dose. Not necessarily the duration (number of minutes) of each session, but the total number of sessions a participant has received. Gold has done some work in this area. Is there is a certain number of sessions that are needed to reach a therapeutic outcome? The number of sessions/week is good, but the number of total sessions is important.

Response: We have added the subgroup analysis by total sessions a participant has received according to this comment. 

11. Table 1 has the mean age. I recommend including the SD as well.

Response: We have added the SD in table 1 

Thank you for taking the time to consider these suggestions. While receiving critical feedback can be difficult, please understand that my intentions are to improve the paper and ensure it has maximum impact. This is an important addition to the literature and I am grateful to the authors for their scholarship. I wish you the best!

Response: Thanks very much for your important comments, these comments are all valuable and very helpful for revising and improving our paper, as well as the important guiding significance to our researches. 

Reviewer #2: This article addresses an important topic that is of interest to music therapists, psychiatrists and teachers and metal health practitioners. The statistics look promising. However, the major concern is that the definition of music therapy is theoretically and practically incorrect and misleading:

"7 Music therapy was defined as music therapy provided by a qualified music teacher, psychological

8 therapist, or nurse. " The study is missing several research studies that I am aware of and this makes its content suspicious. Also missing is a more depth-ful analysis of what active and passive music therapy is, and if it is indeed performed by those in other professions who have no training in 'musuc therapy;'-than the contents and findings are misleading and irrelevant.

Response: (1) We have amendded the difinition of music therapy. According to the American Music Therapy Association (AMTA), “music therapy is the clinical and evidence-based use of music interventions to accomplish individualized goals within a therapeutic relationship by a credentialed professional who has completed an approved music therapy program”.. [American Music Therapy Association (2020). Definition and Quotes about Music Therapy. Available online at: https://www.musictherapy.org/about/quotes/ (Accessed Sep 13, 2020).] 

(2)We are very sorry for missing several research studies in our present meta-analysis. According to this comment, we have performed more extensive electronic search using the following terms: depression or mood disorders or affective disorders and music. We also performed manual search for the reference of all relevent reviews. In order to ensure the study quality of included papers, we excluded the studies with a very small sample size (n<20), we also excluded the non-english papers due to our language barrier. We included 23 new papers and deleted 1 old paper, in the last a total of 55 paper were included in our present analysis. The following are the new included papers and some excluded papers: 

New-included papers

1)Albornoz Y. The effects of group improvisational music therapy on depression in adolescents and adults with substance abuse: a randomised controlled trial. Nordic Journal of Music Therapy 2011;20(3):208–24.

2)Hendricks CB, Robinson B, Bradley B, Davis K. Using music techniques to treat adolescent depression. Journal of Humanistic Counseling, Education and Development 1999; 38:39–46. （unavaliable）

3)Hendricks CB. A study of the use of music therapy techniques in a group for the treatment of adolescent depression. Dissertation Abstracts International 2001;62(2-A):472.

4)Radulovic R. The using of music therapy in treatment of depressive disorders. Summary of Master Thesis. Belgrade: Faculty of Medicine University of Belgrade, 1996. 

5)Zerhusen JD, Boyle K, Wilson W. Out of the darkness: group cognitive therapy for depressed elderly. Journal of Military Nursing Research 1995;1:28–32. PUBMED: 1941727]

6)Chen SC, Yeh ML, Chang HJ, Lin MF. Music, heart rate variability, and symptom clusters: a comparative study. Support Care Cancer. 2020;28(1):351-360. doi:10.1007/s00520-019-04817-x

7)Chang, M. Y., Chen, C. H., and Huang, K. F. (2008). Effects of music therapy on psychological health of women during pregnancy. J. Clin. Nurs. 17, 2580–2587. doi: 10.1111/j.1365-2702.2007.02064.x

8)Chen XJ, Hannibal N, Gold C. Randomized Trial of Group Music Therapy With Chinese Prisoners: Impact on Anxiety, Depression, and Self-Esteem. Int J Offender Ther Comp Criminol. 2016;60(9):1064-1081. doi:10.1177/0306624X15572795

9)Esfandiari, N., and Mansouri, S. (2014). The effect of listening to light and heavy music on reducing the symptoms of depression among female students. Arts Psychother. 41, 211–213. doi: 0.1016/j.aip.2014.02.001 

10)Fancourt, D., Perkins, R., Ascenso, S., Carvalho, L. A., Steptoe, A., and Williamon, A. (2016). Effects of group drumming interventions on anxiety, depression, social resilience and inflammatory immune response among mental health service users. PLoS ONE 11:e0151136. doi: 10.1371/journal.pone.0151136

11)Giovagnoli AR, Manfredi V, Parente A, Schifano L, Oliveri S, Avanzini G. Cognitive training in Alzheimer's disease: a controlled randomized study. Neurol Sci. 2017;38(8):1485-1493. doi:10.1007/s10072-017-3003-9

12)Harmat, L., Takács, J., and Bodizs, R. (2008). Music improves sleep quality in students. J. Adv. Nurs. 62, 327–335. doi: 10.1111/j.1365-2648.2008.04602.x

13)Liao J, Wu Y, Zhao Y, et al. Progressive Muscle Relaxation Combined with Chinese Medicine Five-Element Music on Depression for Cancer Patients: A Randomized Controlled Trial. Chin J Integr Med. 2018;24(5):343-347. doi:10.1007/s11655-017-2956-0

14)Lu, S. F., Lo, C. H. K., Sung, H. C., Hsieh, T. C., Yu, S. C., and Chang, S. C. (2013). Effects of group music intervention on psychiatric symptoms and depression in patient with schizophrenia. Complement. Ther. Med. 21, 682–688. doi: 10.1016/j.ctim.2013.09.002

15)Mahendran R, Gandhi M, Moorakonda RB, et al. Art therapy is associated with sustained improvement in cognitive function in the elderly with mild neurocognitive disorder: findings from a pilot randomized controlled trial for art therapy and music reminiscence activity versus usual care. Trials. 2018;19(1):615. Published 2018 Nov 9. doi:10.1186/s13063-018-2988-6

16)Nwebube C, Glover V, Stewart L. Prenatal listening to songs composed for pregnancy and symptoms of anxiety and depression: a pilot study. BMC Complement Altern Med. 2017;17(1):256. Published 2017 May 8. doi:10.1186/s12906-017-1759-3

17)Porter S, McConnell T, McLaughlin K, et al. Music therapy for children and adolescents with behavioural and emotional problems: a randomised controlled trial. J Child Psychol Psychiatry. 2017;58(5):586-594. doi:10.1111/jcpp.12656

18)Raglio A, Giovanazzi E, Pain D, et al. Active music therapy approach in amyotrophic lateral sclerosis: a randomized-controlled trial. Int J Rehabil Res. 2016;39(4):365-367. doi:10.1097/MRR.0000000000000187

19)Torres E, Pedersen IN, Pérez-Fernández JI. Randomized Trial of a Group Music and Imagery Method (GrpMI) for Women with Fibromyalgia. J Music Ther. 2018;55(2):186-220. doi:10.1093/jmt/thy005

20)Verrusio, W., Andreozzi, P., Marigliano, B., Renzi, A., Gianturco, V., Pecci, M. T., et al. (2014). Exercise training and music therapy in elderly with depressive syndrome: a pilot study. Complement. Ther. Med. 22, 614–620. doi: 10.1016/j.ctim.2014.05.012

21)Wang, J. , Wang, H. and Zhang, D. (2011) Impact of group music therapy on the depression mood of college students. Health, 3, 151-155 

22)Yap AF, Kwan YH, Tan CS, Ibrahim S, Ang SB. Rhythm-centred music making in community living elderly: a randomized pilot study. BMC Complement Altern Med. 2017 Jun 14;17(1):311. doi: 10.1186/s12906-017-1825-x. PMID: 28615007; PMCID: PMC5470187.

23)Koelsch, S., Offermanns, K., and Franzke, P. (2010). Music in the treatment of affective disorders: an exploratory investigation of a new method for music-therapeutic research. Music Percept. Interdisc. J. 27, 307–316. doi: 10.1525/mp.2010.27.4.307 

Excluded papers：

24)Bally, K., Campbell, D., Chesnick, K., and Tranmer, J. E. (2003). Effects of patient controlled music therapy during coronary angiography on procedural pain and anxiety distress syndrome. Crit. Care Nurse 23, 50–58. （not provide useable data）

25)Atiwannapat P, Thaipisuttikul P, Poopityastaporn P, Katekaew W. Active versus receptive group music therapy for major depressive disorder - a pilot study. Complementary Therapies in Medicine 2016;26:141–5. (sample size<20)

26)Garrido S, Stevens CJ, Chang E, Dunne L, Perz J. Music and Dementia: Individual Differences in Response to Personalized Playlists. J Alzheimers Dis. 2018;64(3):933-941. doi:10.3233/JAD-180084 （not randomised or quasi-randomised controlled trials）

27)Sánchez A, Maseda A, Marante-Moar MP, de Labra C, Lorenzo-López L, Millán-Calenti JC. Comparing the Effects of Multisensory Stimulation and Individualized Music Sessions on Elderly People with Severe Dementia: A Randomized Controlled Trial. J Alzheimers Dis. 2016;52(1):303-315. doi:10.3233/JAD-151150 （the control group also received music intervention）

28)Mondanaro JF, Homel P, Lonner B, Shepp J, Lichtensztein M, Loewy JV. Music Therapy Increases Comfort and Reduces Pain in Patients Recovering From Spine Surgery. Am J Orthop (Belle Mead NJ). 2017;46(1):E13-E22. (No full text available)

29)Castillo-Pérez, S., Gómez-Pérez, V., Velasco, M. C., Pérez-Campos, E., and Mayoral, M. A. (2010). Effects of music therapy on depression compared with psychotherapy. Arts Psychother. 37, 387–390. doi: 0.1016/j.aip.2010.07.001 （not provide useable data)

30)Alcântara-Silva TR, de Freitas-Junior R, Freitas NMA, et al. Music Therapy Reduces Radiotherapy-Induced Fatigue in Patients With Breast or Gynecological Cancer: A Randomized Trial. Integr Cancer Ther. 2018;17(3):628-635. doi:10.1177/1534735418757349（not provide useable data)

31)Cheung CWC, Yee AWW, Chan PS, et al. The impact of music therapy on pain and stress reduction during oocyte retrieval - a randomized controlled trial. Reprod Biomed Online. 2018;37(2):145-152. doi:10.1016/j.rbmo.2018.04.049（not provide useable data)

32)Pezzin LE, Larson ER, Lorber W, McGinley EL, Dillingham TR. Music-instruction intervention for treatment of post-traumatic stress disorder: a randomized pilot study. BMC Psychol. 2018;6(1):60. Published 2018 Dec 19. doi:10.1186/s40359-018-0274-8 (the control group also received music intervention)

33)Silverman, M. J. (2011). Effects of music therapy on change and depression on clients in detoxification. J. Addict. Nurs. 22, 185–192. doi: 10.3109/10884602.2011.616606 （the control group also received music intervention）

34)Särkämö T, Laitinen S, Numminen A, Kurki M, Johnson JK, Rantanen P. Clinical and Demographic Factors Associated with the Cognitive and Emotional Efficacy of Regular Musical Activities in Dementia. J Alzheimers Dis. 2016;49(3):767-81. doi: 10.3233/JAD-150453. PMID: 26519435.

35)Tuinmann G, Preissler P, Böhmer H, Suling A, Bokemeyer C. The effects of music therapy in patients with high-dose chemotherapy and stem cell support: a randomized pilot study. Psychooncology. 2017 Mar;26(3):377-384. doi: 10.1002/pon.4142. Epub 2016 May 5. PMID: 27146798.（not provide useable data)

36)Hsu, W. C., and Lai, H. L. (2004). Effects of music on major depression in psychiatric inpatients. Arch. Psychiat. Nurs. 18, 193–199. doi: 10.1016/j.apnu.2004.07.007（not provide useable data)

（3）We have added some new analyses of our data. 1) including three new papers and re-analying of our data, 2) adding the comparison of music therapy and music medicine (figure 3 in revised manuscript) , 3) adding some subgroup analyses by country having music therapy profession, intervention settings, therapy mode, specific music therapy methord, intervenor /therapist, and total intervention session (table 2 in revised manuscript) . 

Reviewer #3: I only reviewed the statistical approach used in this paper, which appeared appropriate for the research question under study. There are two caveats:

1. The authors state that they excluded studies with fewer than 20 participants in one place in the paper (page 4), but fewer than 30 participants in another place in the paper (Table 4). This needs to be corrected for consistency.

Response: We are sorry for making this mistake. In the Methord section, we defined exclusive criteria as studies with a very small sample size (n<20),while in table4 we performed the sensitivity analyses by excluding the papers with smale sample size ( 20< n<30). We have amended the table 4. 

2. The authors mention stronger effects of short and medium length vs. long music therapy periods in their results but there is no accompanying figure. I think it would be beneficial to show these findings in a figure (Forest plot).

Response: We have added these findings with a forest plot (figure 6 in revised manuscript) according to the comment. 

6. PLOS authors have the option to publish the peer review history of their article (what does this mean?). If published, this will include your full peer review and any attached files.

Do you want your identity to be public for this peer review? For information about this choice, including consent withdrawal, please see our Privacy Policy.

Reviewer #1: No

Reviewer #2: No

Reviewer #3: No

---

## [Editor Report · Decision Letter 1]

5 Oct 2020

Effects of music therapy on depression: a meta-analysis of randomized controlled trials

PONE-D-20-17706R1

Dear Dr. Ye,

We’re pleased to inform you that your manuscript has been judged scientifically suitable for publication and will be formally accepted for publication once it meets all outstanding technical requirements.

Kind regards,

Sukru Torun

Academic Editor

PLOS ONE

Additional Editor Comments (optional):

Thank you for your satisfactory effort to correct the many important issues highlighted in my and our reviewers' comments, and to make necessary adjustments that meet our recommendations to improve the quality of your article.
---

## [Editor Report · Acceptance letter]

30 Oct 2020

PONE-D-20-17706R1 

Effects of music therapy on depression: a meta-analysis of randomized controlled trials 

Dear Dr. Ye:

I'm pleased to inform you that your manuscript has been deemed suitable for publication in PLOS ONE. Congratulations! Your manuscript is now with our production department. 

Kind regards, 

on behalf of

Prof. Dr. Sukru Torun 

Academic Editor

PLOS ONE